# Coresets for $k$-Segmentation of Streaming Data

**Guy Rosman** [*][†]
CSAIL, MIT
32 Vassar St., 02139,
Cambridge, MA USA
rosman@csail.mit.edu

**Mikhail Volkov** [†]
CSAIL, MIT
32 Vassar St., 02139,
Cambridge, MA USA
mikhail@csail.mit.edu

**Danny Feldman** [†]
CSAIL, MIT
32 Vassar St., 02139,
Cambridge, MA USA
dannyf@csail.mit.edu

**John W. Fisher III**
CSAIL, MIT
32 Vassar St., 02139,
Cambridge, MA USA
fisher@csail.mit.edu

**Daniela Rus** [†]
CSAIL, MIT
32 Vassar St., 02139,
Cambridge, MA USA
rus@csail.mit.edu

## Abstract

Life-logging video streams, financial time series, and Twitter tweets are a few examples of high-dimensional signals over practically unbounded time. We consider the problem of computing optimal segmentation of such signals by a $k$-piecewise linear function, using only one pass over the data by maintaining a *coreset* for the signal. The coreset enables fast further analysis such as automatic summarization and analysis of such signals.

A coreset (core-set) is a compact representation of the data seen so far, which approximates the data well for a specific task – in our case, segmentation of the stream. We show that, perhaps surprisingly, the segmentation problem admits coresets of cardinality only linear in the number of segments $k$, independently of both the dimension $d$ of the signal, and its number $n$ of points. More precisely, we construct a representation of size $O(k \log n / \varepsilon^2)$ that provides a $(1+\varepsilon)$-approximation for the sum of squared distances to any given $k$-piecewise linear function. Moreover, such coresets can be constructed in a parallel streaming approach. Our results rely on a novel reduction of statistical estimations to problems in computational geometry. We empirically evaluate our algorithms on very large synthetic and real data sets from GPS, video and financial domains, using 255 machines in Amazon cloud.

## 1 Introduction

There is an increasing demand for systems that learn long-term, high-dimensional data streams. Examples include video streams from wearable cameras, mobile sensors, GPS, financial data and biological signals. In each, a time instance is represented as a high-dimensional feature, for example location vectors, stock prices, or image content feature histograms.

We develop real-time algorithms for summarization and segmentation of large streams, by compressing the signals into a compact meaningful representation. This representation can then be used to enable fast analyses such as summarization, state estimation and prediction. The proposed algorithms support data streams that are too large to store in memory, afford easy parallelization, and are generic in that they apply to different data types and analyses. For example, the summarization of wearable video data can be used to efficiently detect different scenes and important events, while collecting GPS data for citywide drivers can be used to learn weekly transportation patterns and characterize driver behavior.

---

[*] Guy Rosman was partially supported by MIT-Technion fellowship

[†] Support for this research has been provided by Hon Hai/Foxconn Technology Group and MIT Lincoln Laboratory. The authors are grateful for this support.

In this paper we use a data reduction technique called *coresets* [1, 9] to enable rapid content-based segmentation of data streams. Informally, a coreset $D$ is *problem dependent* compression of the original data $P$, such that running algorithm $A$ on the coreset $D$ yields a result $A(D)$ that *provably* approximates the result $A(P)$ of running the algorithm on the original data. If the coreset $D$ is small and its construction is fast, then computing $A(D)$ is fast even if computing the result $A(P)$ on the original data is intractable. See definition 2 for the specific coreset which we develop in this paper.

## 1.1 Main Contribution

The main contributions of the paper are: (i) A new coreset for the $k$-segmentation problem (as given in Subsection 1.2) that can be computed at one pass over streaming data (with $O(\log n)$ insertion time/space) and supports distributed computation. Unlike previous results, the insertion time per new observation and required memory is only linear in both the dimension of the data, and the number $k$ of segments. This result is summarized in Theorem 4, and proven in the supplementary material. Our algorithm is scalable, parallelizable, and provides a provable approximation of the cost function. (ii) Using this novel coreset we demonstrate a new system for segmentation and compression of streaming data. Our approach allows realtime summarization of large-scale video streams in a way that preserves the semantic content of the aggregated video sequences, and is easily extendable. (iii) Experiments to demonstrate our approach on various data types: video, GPS, and financial data. We evaluate performance with respect to output size, running time and quality and compare our coresets to uniform and random sample compression. We demonstrate the scalability of our algorithm by running our system on an Amazon cluster with 255 machines with near-perfect parallelism as demonstrated on $256,000$ frames. We also demonstrate the effectiveness of our algorithm by running several analysis algorithms on the computed coreset instead of the full data. Our implementation summarizes the video in less than 20 minutes, and allows real-time segmentation of video streams at 30 frames per second on a single machine.

**Streaming and Parallel computations.** Maybe the most important property of coresets is that even an efficient off-line construction implies a fast construction that can be computed (a) Embarrassingly in parallel (e.g. cloud and GPUs), (b) in the streaming model where the algorithm passes only once over the (possibly unbounded) streaming data. Only small amount of memory and update time ($\sim \log n$) per new point insertion is allowed, where $n$ is the number of observations so far.

## 1.2 Problem Statement

The $k$-segment mean problem optimally fits a given discrete time signal of $n$ points by a set of $k$ linear segments over time, where $k \geq 1$ is a given integer. That is, we wish to partition the signal into $k$ consecutive time intervals such that the points in each time interval are lying on a single line; see Fig. 1(left) and the following formal definition.

We make the following assumptions with respect to the data: (a) We assume the data is represented by a feature space that suitably represents its underlying structure; (b) The content of the data includes at most $k$ segments that we wish to detect automatically; An example for this are scenes in a video, phases in the market as seen by stock behaviour, etc. and (c) The dimensionality of the feature space is often quite large (from tens to thousands of features), with the specific choice of the features being application dependent – several examples are given in Section 3. This motivates the following problem definition.

**Definition 1** ($k$-segment mean). *A set $P$ in $\mathbb{R}^{d+1}$ is a* signal *if* $P = \{(1, p_1), (2, p_2), \cdots, (n, p_n)\}$ *where $p_i \in \mathbb{R}^d$ is the point at time index $i$ for every $i = [n] = \{1, \cdots, n\}$. For an integer $k \geq 1$, a $k$-*segment is a $k$-piecewise linear function $f : \mathbb{R} \to \mathbb{R}^d$ that maps every time $i \in \mathbb{R}$ to a point $f(i)$ in $\mathbb{R}^d$. The* fitting *error at time $t$ is the squared distance between $p_i$ and its corresponding projected point $f(i)$ on the $k$-segments. The* fitting *cost of $f$ to $P$ is the sum of these squared distances,*

$$\text{cost}(P, f) = \sum_{i=1}^{n} \|p_i - f(i)\|_2^2, \tag{1}$$

*where $\| \cdot \|$ denotes the Euclidean distance. The function $f$ is a $k$-*segment mean *of $P$ if it minimizes* $\text{cost}(P, f)$.

For the case $k = 1$ the 1-segment mean is the solution to the linear regression problem. If we restrict each of the $k$-segments to be a horizontal segment, then each segment will be the mean height of the corresponding input points. The resulting problem is similar to the $k$-mean problem, except

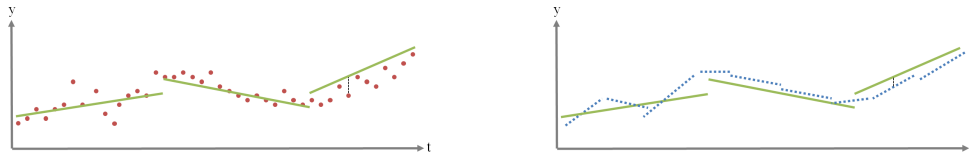

Figure 1: For every $k$-segment $f$, the cost of input points (red) is approximated by the cost of the coreset (dashed blue lines). Left: An input signal and a 3-segment $f$ (green), along with the regression distance to one point (dashed black vertical lines). The cost of $f$ is the sum of these squared distances from all the input points. Right: The coreset consists of the projection of the input onto few segments, with approximate per-segment representation of the data.

each of the voronoi cells is forced to be a single region in time, instead of nearest center assignment, i.e. the regions are contiguous.

In this paper we are interested in seeking a compact representation $D$ that approximates $\mathrm{cost}(P, f)$ for every $k$-segment $f$ using the above definition of $\mathrm{cost}'(D, f)$. We denote a set $D$ as a $(k, \varepsilon)$-coreset according to the following definition,

**Definition 2** $((k, \varepsilon)$-coreset$)$. *Let $P \subseteq \mathbb{R}^{d+1}$, $k \geq 1$ be an integer, for some small $\varepsilon > 0$. A set $D$, with a cost function $\mathrm{cost}'(\cdot)$ is a $(k, \varepsilon)$-coreset for $P$ if for every $k$-segment $f$ we have*

$$(1 - \varepsilon)\mathrm{cost}(P, f) \leq \mathrm{cost}'(D, f) \leq (1 + \varepsilon)\mathrm{cost}(P, f).$$

We present a new coreset construction with provable approximations for a family of natural $k$-segmentation optimization problems. This is the first such construction whose running time is linear in both the number of data points $n$, their dimensionality $d$, and the number $k$ of desired segments. The resulting coreset consists of $O(dk/\varepsilon^2)$ points that approximates the sum of square distances for *any* $k$-piecewise linear function ($k$ segments over time). In particular, we can use this coreset to compute the $k$-piecewise linear function that minimize the sum of squared distances to the input points, given arbitrary constraints or weights (priors) on the desired segmentation. Such a generalization is useful, for example, when we are already given a set of candidate segments (e.g. maps or distribution of images) and wish to choose the right $k$ segments that approximate the input signal.

Previous results on coresets for $k$-segmentation achieved running time or coreset size that are at least quadratic in $d$ and cubic in $k$ [12, 11]. As such, they can be used with very large data, for example to long streaming video data which is usually high-dimensional and contains large number of scenes. This prior work is based on some non-uniform sampling of the input data. In order to achieve our results, we had to replace the sampling approach by a new set of deterministic algorithms that carefully select the coreset points.

## 1.3 Related Work

Our work builds on several important contributions in coresets, $k$-segmentations, and video summarization.

**Approximation Algorithms.** One of the main challenges in providing provable guarantees for segmentation w.r.t segmentation size and quality is global optimization. Current provable algorithms for data segmentation are cubic-time in the number of desired segments, quadratic in the dimension of the signal, and cannot handle both parallel and streaming computation as desired for big data. The closest work that provides provable approximations is that of [12].

Several works attempt to summarize high-dimensional data streams in various application domains. For example, [19] describe the video stream as a high-dimensional stream and run approximated clustering algorithms such as $k$-center on the points of the stream; see [14] for surveys on stream summarization in robotics. The resulting $k$-centers of the clusters comprise the video summarization. The main disadvantages of these techniques are (i) They partition the data stream into $k$ clusters that do not provide $k$-segmentation over time. (ii) Computing the $k$-center takes time exponential in both $d$ and $k$ [16]. In [19] heuristics were used for dimension reduction, and in [14] a 2-approximation was suggested for the off-line case, which was replaced by a heuristic forstreaming. (iii) In the context of analysis of video streams, they use a feature space that is often simplistic and does not utilize the large data available effciently. In our work the feature space can be updated on-line using a coreset for $k$-means clustering of the features seen so far.

**$k$-segment Mean.** The $k$-segment mean problem can be solved exactly using dynamic programming [4]. However, this takes $O(dn^2 k)$ time and $O(dn^2)$ memory, which is impractical for streaming data. In [15, Theorem 8] a $(1 + \varepsilon)$-approximation was suggested using $O(n(dk)^4 \log n/\varepsilon)$ time. While

the algorithm in [15] support efficient streaming, it is not parallel. Since it returns a $k$-segmentation and not a coreset, it cannot be used to solve other optimization problems with additional priors or constraints. In [12] an improved algorithm that takes $O(nd^2k + ndk^3)$ time was suggested. The algorithm is based on a coreset of size $O(dk^3/\varepsilon^3)$. Unlike the coreset in this paper, the running time of [12] is cubic in both $d$ and $k$. The result in [12] is the last in a line of research for the $k$-segment mean problem and its variations; see survey in [11, 15, 13]. The application was segmentation of 3-dimensional GPS signal (time, latitude, longitude). The coreset construction in [12] and previous papers takes time and memory that is quadratic in the dimension $d$ and cubic in the number of segments $k$. Conversely, our coreset construction takes time only linear in both $k$ and $d$. While recent results suggest running time linear in $n$, and space that is near-logarithmic in $n$, the computation time is still cubic in $k$, the number of segments, and quadratic in $d$, the dimension. Since the number $k$ represents the number of scenes, and $d$ is the feature dimensionality, this complexity is prohibitive.

**Video Summarization** One motivating application for us is online video summarization, where input video stream can be represented by a set of points over time in an appropriate feature space. Every point in the feature space represents the frame, and we aim to produce a compact approximation of the video in terms of this space and its Euclidean norm. Application-aware summarization and analysis of ad-hoc video streams is a difficult task with many attempts aimed at tackling it from various perspectives [5, 18, 2]. The problem is highly related to video action classification, scene classification, and object segmentation [18]. Applications where life-long video stream analysis is crucial include mapping and navigation medical / assistive interaction, and augmented-reality applications, among others. Our goal differs from video compression in that compression is geared towards preserving image quality for all frames, and therefore stores semantically redundant content. Instead, we seek a summarization approach that allows us to represent the video content by a set of key segments, for a given feature space.

This paper is organized as follows. We begin by describing the $k$-segmentation problem and the proposed coresets, and describe their construction, and their properties in Section 2. We perform several experiments in order to validate the proposed approach on data collected from GPS and werable web-cameras, and demonstrate the aggregation and analysis of multiple long sequences of wearable user video in Section 3. Section 4 concludes the paper and discusses future directions.

## 2   A Novel Coreset for $k$-segment Mean

The key insights for constructing the $k$-segment coreset are: i) We observe that for the case $k = 1$, a 1-segment coreset can be easily obtained using SVD. ii) For the general case, $k \geq 2$ we can partition the signal into a suitable number of intervals, and compute a 1-segment coreset for each such interval. If the number of intervals and their lengths are carefully chosen, most of them will be well approximated by every $k$-segmentation, and the remaining intervals will not incur a large error contribution.

Based on these observations, we propose the following construction. 1) Estimate the signal's complexity, i.e., the approximated fitting cost to its $k$-segment mean. We denote this step as a call to the algorithm BICRITERIA. 2) Given an complexity measure for the data, approximate the data by a set of segments with auxiliary information, which is the proposed coreset, denoted as the output of algorithm BALANCEDPARTITION.

We then prove that the resulting coreset allows us to approximate with guarantees the fitting cost for any $k$-segmentation over the data, as well as compute an optimal $k$-segmentation. We state the main result in Theorem 4, and describe the proposed algorithms as Algorithms 1 and 2. We refer the reader to the supplementary material for further details and proofs.

### 2.1   Computing a $k$-Segment Coreset

We would like to compute a $(k, \varepsilon)$-coreset for our data. A $(k, \varepsilon)$-coreset $D$ for a set $P$ approximates the fitting cost of any query $k$-segment to $P$ up to a small multiplicative error of $1 \pm \varepsilon$. We note that a $(1, 0)$-coreset can be computed using SVD; See the supplementary material for details and proof. However, for $k > 2$, we cannot approximate the data by a representative point set (we prove this in the supplementary material). Instead, we define a data structure $D$ as our proposed coreset, and define a new cost function $\text{cost}'(D, f)$ that approximates the cost of $P$ to any $k$-segment $f$.

The set $D$ consists of tuples of the type $(C, g, b, e)$. Each tuple corresponds to a different time interval $[b, e]$ in $\mathbb{R}$ and represents the set $P(b, e)$ of points in this interval. $g$ is the 1-segment mean of the data $P$ in the interval $[b, e]$. The set $C$ is a $(1, \varepsilon)$-coreset for $P(b, e)$.

We note the following: 1) If all the points of the $k$-segment $f$ are on the same segment in this time interval, i.e, $\{f(t) \mid b \leq t \leq e\}$ is a linear segment, then the cost from $P(b, e)$ to $f$ can be approximated well by $C$, up to $(1 + \varepsilon)$ multiplicative error. 2) If we project the points of $P(b, e)$ on their 1-segment mean $g$, then the projected set $L$ of points will approximate well the cost of $P(b, e)$ to $f$, even if $f$ corresponds to more than one segment in the time interval $[b, e]$. Unlike the previous case, the error here is additive. 3) Since $f$ is a $k$-segment there will be at most $k - 1$ time intervals that will intersect more than two segments of $f$, so the overall additive error is small . This motivates the following definition of $D$ and $\mathrm{cost}'$.

**Definition 3** ($\mathrm{cost}'(D, f)$). *Let $D = \{(C_i, g_i, b_i, e_i)\}_{i=1}^{m}$ where for every $i \in [m]$ we have $C_i \subseteq \mathbb{R}^{d+1}$, $g_i : \mathbb{R} \to \mathbb{R}^d$ and $b_i \leq e_i \in \mathbb{R}$. For a $k$-segment $f : \mathbb{R} \to \mathbb{R}^d$ and $i \in [m]$ we say that $C_i$ is* served by one segment *of $f$ if $\{f(t) \mid b_i \leq t \leq e_i\}$ is a linear segment. We denote by $\mathrm{Good}(D, f) \subseteq [m]$ the union of indexes $i$ such that $C_i$ is served by one segment of $f$. We also define $L_i = \{g_i(t) \mid b_i \leq t \leq e_i\}$, the projection of $C_i$ on $g_i$. We define $\mathrm{cost}'(D, f)$ as $\sum_{i \in \mathrm{Good}(D, f)} \mathrm{cost}(C_i, f) + \sum_{i \in [m] \setminus \mathrm{Good}(D, f)} \mathrm{cost}(L_i, f)$.*

Our coreset construction for general $k > 1$ is based on an input parameter $\sigma > 0$ such that for an appropriate $\sigma$ the output is a $(k, \varepsilon)$-coreset. $\sigma$ characterizes the complexity of the approximation. The BICRITERIA algorithm, given as Algorithm 1, provides us with such an approximation. Properties of this algorithms are described in the supplementary material.

**Theorem 4.** *Let $P = \{(1, p_1), \cdots, (n, p_n)\}$ such that $p_i \in \mathbb{R}^d$ for every $i \in [n]$. Let $D$ be the output of a call to* BALANCEDPARTITION$(P, \varepsilon, \sigma)$*, and let $f$ be the output of* BICRITERIA$(P, k)$*; Let $\sigma = \mathrm{cost}(f)$. Then $D$ is a $(k, \varepsilon)$-coreset for $P$ of size $|D| = O(k) \cdot \left(\log n / \varepsilon^2\right)$, and can be computed in $O(dn / \varepsilon^4)$ time.*

*Proof.* We give a sketch of the proof, also given in Theorem 10 in the supplementary material, and accompanying theorems. Lemma 8 states that given an estimate $\sigma$ of the optimal segmentation cost, BALANCEDPARTITION$(P, \varepsilon, \sigma)$ provides a $(k, \varepsilon)$-coreset of the data $P$. This hinges on the observation that given a fine enough segmentation of the time domain, for each segment we can approximate the data by an SVD with bounded error. This approximation is exact for $1 - segments$ (See claim 2 in the supplementary material), and can be bounded for a $k$-segments because of the number of segment intersections. According to Theorem 9 of the supplementary material, $\sigma$ as computed by BICRITERIA$(P, k)$ provides such an approximation. $\square$

---

**Algorithm 1:** BICRITERIA$(P, k)$

---

**Input**: A set $P \subseteq \mathbb{R}^{d+1}$ and an integer $k \geq 1$
**Output**: A bicriteria $(O(\log n), O(\log n))$-approximation to the $k$-segment mean of $P$.

1  **if** $n \leq 2k + 1$ **then**
2  $\quad$ $f :=$ a 1-segment mean of $P$;
3  $\quad$ **return** $f$;

4  Set $t_1 \leq \cdots \leq t_n$ and $p_1, \cdots, p_n \in \mathbb{R}^d$ such that $P = \{(t_1, p_1), \cdots, (t_n, p_n)\}$
5  $m \leftarrow \{t \in \mathbb{R} \mid (t, p) \in P\}$
6  Partition $P$ into $4k$ sets $P_1, \cdots, P_{2k} \subseteq P$ such that for every $i \in [2k - 1]$:
$\quad$ (i) $|\{t \mid (t, p) \in P_i\}| = \left\lfloor \dfrac{m}{4k} \right\rfloor$, and $\quad$ (ii) if $(t, p) \in P_i$ and $(t', p') \in P_{i+1}$ then $t < t'$.

7  **for** $i := 1$ *to* $4k$ **do**
8  $\quad$ Compute a 2-approximation $g_i$ to the 1-segment mean of $P_i$

9  $Q :=$ the union of $k + 1$ signals $P_i$ with the smallest value $\mathrm{cost}(P_i, g_i)$ among $i \in [2k]$.
10  $h :=$ BICRITERIA$(P \setminus Q, k)$; Repartition the segments that do not have a good approx.
11  Set
$$f(t) := \begin{cases} g_i(t) & \exists (t, p) \in P_i \text{ such that } P_i \subseteq Q \\ h(t) & \text{otherwise} \end{cases}.$$

12  **return** $f$;

---

---

**Algorithm 2:** BALANCEDPARTITION$(P, \varepsilon, \sigma)$

---

**Input**: A set $P = \{(1, p_1), \cdots, (n, p_n)\}$ in $\mathbb{R}^{d+1}$
an error parameters $\varepsilon \in (0, 1/10)$ and $\sigma > 0$.
**Output**: A set $D$ that satisfies Theorem 4.

1  $Q := \emptyset$; $D := \emptyset$ ; $p_{n+1} :=$ an arbitrary point in $\mathbb{R}^d$ ;
2  **for** $i := 1$ *to* $n + 1$ **do**
3  $\quad$ $Q := Q \cup \{(i, p_i)\}$; Add new point to tuple
4  $\quad$ $f^* :=$ a linear approximation of $Q$; $\quad \lambda := \text{cost}(Q, f^*)$
5  $\quad$ **if** $\lambda > \sigma$ **or** $i = n + 1$ **then**
6  $\quad \quad$ $T := Q \setminus \{(i, p_i)\}$ ; take all the new points into tuple
7  $\quad \quad$ $C :=$ a $(1, \varepsilon/4)$-coreset for $T$; Approximate points by a local representation
8  $\quad \quad$ $g :=$ a linear approximation of $T$, $b := i - |T|$, $e := i - 1$; save endpoints
9  $\quad \quad$ $D := D \cup \{(C, g, b, e)\}$ ; save a tuple
10 $\quad \quad$ $Q := \{(i, p_i)\}$ ; proceed to new point

11 **return** $D$

---

For efficient $k$-segmentation we run a $k$-segment mean algorithm on our small coreset instead of the original large input. Since the coreset is small we can apply dynamic programming (as in [4]) in an efficient manner. In order to compute an $(1 + \varepsilon)$ approximation to the $k$-segment mean of the original signal $P$, it suffices to compute a $(1 + \varepsilon)$ approximation to the $k$-segment mean of the coreset, where cost is replaced by cost$'$. However, since D is not a simple signal, but a more involved data structure, it is not clear how to run existing algorithms on D. In the supplementary material we show how to apply such algorithms on our coresets. In particular, we can run naive dynamic programming [4] on the coreset and get a $(1 + \varepsilon)$ approximate solution in an efficient manner, as we summarize as follows.

**Theorem 5.** *Let $P$ be a d-dimensional signal. A $(1 + \varepsilon)$ approximation to the k-segment mean of $P$ can be computed in $O\left(ndk/\varepsilon + d(klog(n)/\varepsilon)^{O(1)}\right)$ time .*

## 2.2 Parallel and Streaming Implementation

One major advantage of coresets is that they can be constructed in parallel as well as in a streaming setting. The main observation is that the union of coresets is a coreset — if a data set is split into subsets, and we compute a coreset for every subset, then the union of the coresets is a coreset of the whole data set. This allows us to have each machine separately compute a coreset for a part of the data, with a central node which approximately solves the optimization problem; see [10, Theorem 10.1] for more details and a formal proof. As we show in the supplementary material, this allows us to use coresets in the streaming and parallel model.

## 3 Experimental Results

We now demonstrate the results of our algorithm on four data types of varying length and dimensionality. We compare our algorithms against several other segmentation algorithms. We also show that the coreset effectively improves the performance of several segmentation algorithms by running the algorithms on our coreset instead of the full data.

### 3.1 Segmentation of Large Datasets

We first examine the behavior of the algorithm on synthetic data which provides us with easy ground-truth, to evaluate the quality of the approximation, as well as the efficiency, and the scalability of the coreset algorithms. We generate synthetic test data by drawing a discrete $k$-segment $P$ with $k = 20$, and then add Gaussian and salt-and-pepper noise. We then benchmark the computed $(k, \varepsilon)$-coreset $D$ by comparing it against piecewise linear approximations with (1) a uniformly sampled subset of control points $U$ and (2) a randomly placed control points $R$. For a fair comparison between the $(k, \varepsilon)$-coreset $D$ and the corresponding approximations $U, R$ we allow the same number of coefficients for each approximation. Coresets are evaluated by computing the fitting cost to a query $k$-segment $Q$ that is constructed based on the a-priori parameters used to generate $P$.

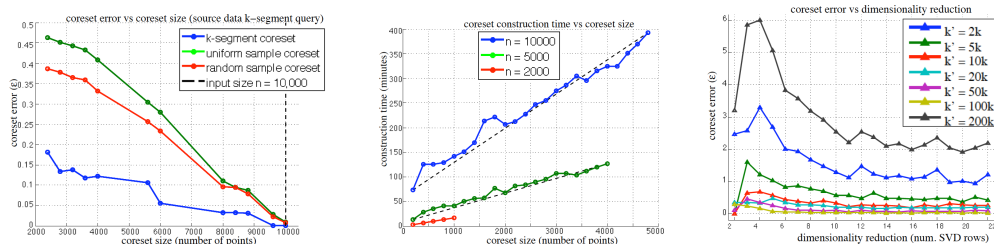

(a) Coreset size vs coreset error     (b) $(k, \varepsilon)$-coreset size vs CPU time     (c) Coreset dim. vs coreset error

Figure 2: Figure 2a shows the coreset error ($\varepsilon$) decreasing as a function of coreset size. The dotted black line indicates the point at which the coreset size is equal to the input size. Figure 2b shows the coreset construction time in minutes as a function of coreset size. Trendlines show the linear increase in construction time with coreset size. Figure 2c shows the reduction in coreset error as a function of the dimensionality of the 1-segment coreset, for fixed input size (dimensionality can often be reduced down to $\mathbb{R}^2$.

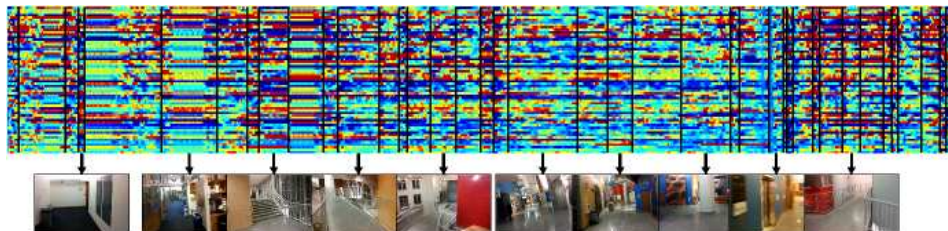

Figure 3: Segmentation from Google Glass. Black vertical lines present segment boundaries, overlayed on top of the bags of word representation. Icon images are taken from the middle of each segment.

**Approximation Power:** Figure 2a shows the aggregated fitting cost error for 1500 experiments on synthetic data. We varied the assumed $k'$ segment complexity. In the plot we show how well a given $k'$ performed as a guess for the true value of $k$. As Figure 2a shows, we significantly outperform the other schemes. As the coreset size approaches the size $P$ the error decreases to zero as expected.

**Coreset Construction Time:** Figure 2b shows the linear relationship between input size and construction time of $D$ for different coreset size. Figure 2c shows how a high dimensionality benefits coreset construction. This is even more apparent in real data which tends to be sparse, so that in practice we are typically able to further reduce the coreset dimension in each segment.

**Scalability:** The coresets presented in this work are parallelizable, as discussed in Section 2.2. We demonstrate scalability by conducting very large scale experiments on both real and synthetic data, running our algorithm on a network of 255 Amazon EC2 vCPU nodes. We compress a 256,000-frame *bags-of-words* (BOW) stream in approximately 20 minutes with almost-perfect scalability. For a comparable single node running on the same data dataset, we estimate a total running time of approximately 42 hours.

### 3.2 Real Data Experiments

We compare our coreset against uniform sample and random sample coresets, as well as two other segmentation techniques: Ramer-Douglas-Peucker (RDP) algorithm [20, 8], and the Dead Reckoning (DR) algorithm [23]. We also show that we can combine our coreset with segmentation algorithms, by running the algorithm on the coresets itself. We emphasize that segmentation techniques were chosen as simple examples and are not intended to reflect the state of the art – but rather to demonstrate how the $k$-segment coreset can improve on any given algorithm.

To demonstrate the general applicability of our techniques, we run our algorithm using financial (1D) time series data, as well as GPS data. For the 1D case we use price data from the Mt.Gox Bitcoin exchange. Bitcoin is of interest because its price has grown exponentially with its popularity in the past two years. Bitcoin has also sustained several well-documented market crashes [3],[6] that we can relate to our analysis. For the 2D case we use GPS data from 343 taxis in San Francisco. This is of interest because a taxi-route segmentation has an intuitive interpretation that we can easily evaluate, and on the other hand GPS data forms an increasingly large information source in which we are interested.

Figure 4a shows the results for Bitcoin data. Price extrema are highlighted by local price highs (green) and lows (red). We observe that running the DR algorithm on our $k$-segment coreset captures these events quite well. Figures 4b,4c show example results for a single taxi. Again, we observe that the DR segmentation produces segments with a meaningful spatial interpretation. Figure 5 shows a plot of coreset errors for the first 50 taxis (right), and the table gives a summary of experimental results for the Bitcoin and GPS experiments.

### 3.3 Semantic Video Segmentation

In addition, we demonstrate use of the proposed coreset for video streams summarization. While different choices of frame representations for video summarization are available [22, 17, 18], we used color-augmented SURF features, quantized into 5000 visual words, trained on the ImageNet 2013 dataset [7]. The resulting histograms are compressed in a streaming coreset. Computation in on a single core runs at 6Hz; A parallel version achieves 30Hz on a single i7 machine, processing 6 hours of video in 4 hours on a single machine, i.e. faster than real-time.

In Figure 3 we demonstrate segmentation of a video taken from Google Glass. We visualize BOWs, as well as the segments suggested by the $k$-segment mean algorithm [4] run on the coreset. Inspecting the results, most segment transitions occur at scene and room changes.

Even though optimal segmentation can not be done in real-time, the proposed coreset is computed in real-time and can further be used to automatically summarize the video by associating representative frames with segments. To evaluate the "semantic" quality of our segmentation, we compared the resulting segments to uniform segmentation by contrasting them with a human annotation of the video into scenes. Our method gave a 25% improvement (in Rand index [21]) over a 3000 frames sequence.

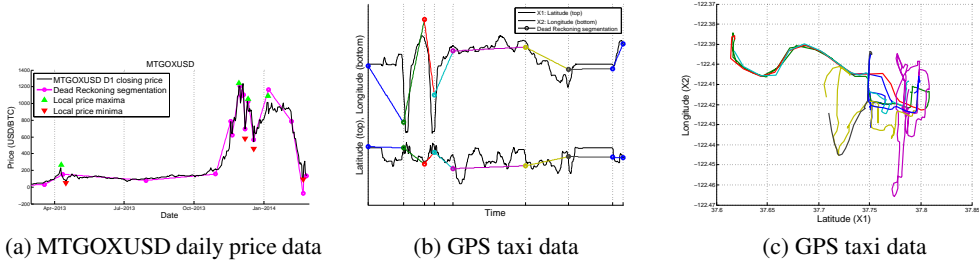

| (a) MTGOXUSD daily price data | (b) GPS taxi data | (c) GPS taxi data |

Figure 4: (a) shows the Bitcoin prices from 2013 on, overlayed with a DR segmentation computed on our coreset. The red/green triangles indicate prominent market events. (b) 4c shows normalized GPS data overlayed with a DR segmentation computed on our coreset. (c) shows a lat/long plot (right) demonstrating that the segmentation yields a meaningful spatial interpretation.

| Average $\varepsilon$ | Bitcoin data | GPS data |
|---|---|---|
| **$k$-segment coreset** | **0.0092** | **0.0014** |
| Uniform sample coreset | 1.8726 | 0.0121 |
| Random sample coreset | 8.0110 | 0.0214 |
| RDP on original data | 0.0366 | 0.0231 |
| **RDP on $k$-segment** | **0.0335** | **0.0051** |
| DeadRec on original data | 0.0851 | 0.0417 |
| **DeadRec on $k$-segment** | **0.0619** | **0.0385** |

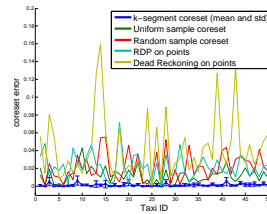

Figure 5: Table: Summary for Bitcoin / GPS data. Plot: Errors / standard deviations for the first 50 cabs.

## 4 Conclusions

In this paper we demonstrated a new framework for segmentation and event summarization of high-dimensional data. We have shown the effectiveness and scalability of the algorithms proposed, and its applicability for large distributed video analysis. In the context of video processing, we demonstrate how using the right framework for analysis and clustering, even relatively straightforward representations of image content lead to a meaningful and reliable segmentation of video streams at real-time speeds.

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
