[Supplementary Material]

# Coresets for k-Segmentation of Streaming Data
## Supplementary Material

Guy Rosman      Mikhail Volkov      Dan Feldman
John W. Fisher III      Daniela Rus

October 30, 2014

## 1   Introduction

In this supplementary material we detail the construction, properties, and proofs for a $k$-segment mean coreset that allows efficient segmentation of high-dimensional signals. We define the $k$-segment mean problem in Section 2. We describe a coreset for the 1-segment mean in Section 3. We show why a similar construction is not possible for the $k$-segment mean problem in Section 4. In Sections 5,6,7 we define the proposed coreset and prove its approximation properties. In Section 8 we extend the coreset so as to allow efficient segmentation by dynamic programming, adapting the classical algorithm of [Bel61]. In Section 9 we discuss the case of streaming and the modifications needed for the algorithm in order to run in streaming mode.

## 2   $k$-Segment mean

Let $P = \{(t_1, p_1), \cdots, (t_n, p_n)\}$ be a subset of $\mathbb{R}^{d+1}$ where $t_i \in \mathbb{R}$ and $p_i \in \mathbb{R}^d$ for every $i \in [n] = \{1, \cdots, n\}$. The *fitting cost* (henceforth simply "cost") from $P$ to a $k$-segment $f$ is the sum of squared distances

$$\mathrm{cost}\,(P, f) = \sum_{(t,p)\in P} \| \,(p - f(t))\,\|^2, \tag{1}$$

where here, as in the paper, $\|X\|^2 = \sum_{ij}(X_{ij})^2$ is the sum of squared entries of a matrix or a vector $X$ (known as the Frobenius norm for a matrix or the $\ell_2$ Euclidean norm for a vector).

A *k-segment mean* of $P$ is a $k$-segment $f^* : \mathbb{R} \to \mathbb{R}^d$ that minimizes $\mathrm{cost}(P, f)$ over every $k$-segment $f : \mathbb{R} \to \mathbb{R}^d$. For $\alpha \geq 1$, an *$\alpha$-approximation for the $k$-segment mean of $P$* is a $k$-segment $f$ such that $\mathrm{cost}(P, f) \leq \alpha \cdot \mathrm{cost}(P, f^*)$. For $\alpha, \beta > 0$, an $(\alpha, \beta)$-approximation for the $k$-segment mean of $P$ is a $(k \cdot \beta)$-segment $g$ such that $\mathrm{cost}(P, g) \leq \alpha \cdot \mathrm{cost}(P, f^*)$.

One of our main tool for computing approximations to the $k$-segment mean is the *singular value decomposition* (SVD) which is defined as follows. For integers $n, d \geq 1$ we denote by $\mathbb{R}^{n \times d}$ the set $n \times d$ matrices whose entries are in $\mathbb{R}$. A *unitary* matrix is a matrix whose columns are orthonormal vectors. The thin SVD of a matrix $X \in \mathbb{R}^{n \times d}$ is $X = U\Sigma V^T$ where both $U \in \mathbb{R}^{n \times d}$ and $V \in \mathbb{R}^{d \times d}$ are unitary matrices, and $\Sigma \in \mathbb{R}^{d \times d}$ is a diagonal matrix of non-negative and non-increasing diagonal entries.

# 3  1-Segment Coreset

A $(1, \varepsilon)$-coreset approximates $\text{cost}(P, f)$ for every 1-segment $f$ up to a factor of $1 \pm \varepsilon$ as defined below.

**Definition 1** $((1, \varepsilon)$-coreset$)$. *Let $P$ and $C$ be two sets in $\mathbb{R}^{d+1}$ and let $\varepsilon, w > 0$. The pair $(C, w)$ is a $(1, \varepsilon)$-coreset for $P$, if for every 1-segment $f : \mathbb{R} \to \mathbb{R}^d$ we have*

$$(1 - \varepsilon)\text{cost}(P, f) \leq w \cdot \text{cost}(C, f) \leq (1 + \varepsilon)\text{cost}(P, f).$$

For example, $(P, w)$ is a $(1, \varepsilon)$-coreset for $P$ with $\varepsilon = 0$ and $w = 1$. However, a coreset is efficient if its size $|C|$ is much smaller than $P$.

It is easy to compute $\text{cost}(P, f)$ exactly by a matrix $\Sigma V^T$ of $(d + 2)$ rows using SVD, as shown in Algorithm 1. In our coreset construction we use additional matrices $Q$ and $Y$ to turn this matrix into a subset $C$ of $\mathbb{R}^{d+1}$ so that the cost $\text{cost}(P, f) = \text{cost}(C, f)$ is still a point-wise cost, although a weighted one. This allows us to improve the result later in this section, to get a less trivial coreset $C$ of only $O(1/\varepsilon^2)$ rows.

---
**Algorithm 1:** 1-SEGMENTCORESET$(P)$

---
**Input**: A set $P = \{(t_1, p_1), \cdots, (t_n, p_n)\}$ in $\mathbb{R}^{d+1}$ .
**Output**: A $(1, 0)$-coreset $(C, w)$ that satisfies Claim 2.
1 Set $X \in \mathbb{R}^{n \times (d+2)}$ to be matrix whose $i$th row is $(1, t_i, p_i)$ for every $i \in [n]$.
2 Compute the thin SVD $X = U \Sigma V^T$ of $X$.
3 Set $u \in \mathbb{R}^{d+2}$ to be the leftmost column of $\Sigma V^T$.
4 Set $w := \frac{\|u\|^2}{d+2}$. /* $w > 0$ since $\|\Sigma\| = \|X\| > 0$                              */
5 Set $Q, Y \in \mathbb{R}^{(d+2) \times (d+2)}$ to be unitary matrices whose leftmost columns are $u/\|u\|$ and $(\sqrt{w}, \cdots, \sqrt{w})/\|u\|$ respectively.
6 Set $B \in \mathbb{R}^{(d+2) \times (d+1)}$ to be the $(d+1)$ rightmost columns of $Y Q^T \Sigma V^T / \sqrt{w}$.
7 Set $C \subseteq \mathbb{R}^{d+1}$ to be the union of the rows in $B$ ;
8 **return** $(C, w)$.

---

**Claim 2.** *Let $P$ be a set of $n$ points in $\mathbb{R}^{d+1}$. Let $(C, w)$ be the output of a call to 1-SEGMENTCORESET$(P)$; see Algorithm 1. Then $(C, w)$ is a $(1, 0)$-coreset for $P$ of size $|C| = d + 1$. Moreover, $C$ and $w$ can be computed in $O(nd^2)$ time.*

*Proof.* Let $f : \mathbb{R} \to \mathbb{R}^d$ be a 1-segment. Hence, there are row vectors $a, b \in \mathbb{R}^d$ such that $f(t) = a + bt$, for every $t \in \mathbb{R}$. By definition of $Q$ and $Y$ we have $Y Q^T u / \|u\| = (\sqrt{w}, \cdots, \sqrt{w})^T / \|u\|$.

The leftmost column of $YQ^T\Sigma V^T$ is thus $YQ^Tu = (\sqrt{w}, \cdots, \sqrt{w})^T$. Therefore,

$$\text{cost}\,(P, f) = \sum_{(t,p)\in P}^{n} \|f(t) - p\|^2 = \sum_{(t,p)\in P} \|a + bt - p\|^2 = \sum_{(t,p)\in P} \left\| \begin{bmatrix} 1 & t \end{bmatrix} \begin{bmatrix} a \\ b \end{bmatrix} - p \right\|^2$$

$$= \left\| X \begin{bmatrix} a \\ b \\ -I \end{bmatrix} \right\|^2 = \left\| U\Sigma V^T \begin{bmatrix} a \\ b \\ -I \end{bmatrix} \right\|^2 = \left\| YQ^T\Sigma V^T \begin{bmatrix} a \\ b \\ -I \end{bmatrix} \right\|^2 = \left\| \begin{bmatrix} \sqrt{w} \\ \vdots \\ \sqrt{w} \end{bmatrix} \quad \sqrt{w}B \end{bmatrix} \begin{bmatrix} a \\ b \\ -I \end{bmatrix} \right\|^2$$

$$= w \left\| \begin{bmatrix} 1 \\ \vdots & B \\ 1 \end{bmatrix} \begin{bmatrix} a \\ b \\ -I \end{bmatrix} \right\|^2 = w \sum_{(t,p)\in B} \|(a + bt - p)\|^2 = w \cdot \text{cost}(C, f).$$

**Construction Time.** The matrices $Q$ and $Y$ can be computed in $O(dn^2)$ time using the QR decomposition of $\begin{bmatrix} (1, \cdots, 1)^T & I \end{bmatrix}$ and $\begin{bmatrix} u & I \end{bmatrix}$. Computing the thin SVD of an $n \times d$ matrix $X$ also takes $O(nd^2)$ time. Hence, the overall running time is $O(nd^2)$ [Pea01]. $\square$

The size $d + 1$ and running time of the above $(1, 0)$-coreset $C$ might be too large, for example when $d$ is in the order of $n$, or we are dealing with high dimensional space such as images or text. On the other side, in the rest of the paper the construction of $(1, \varepsilon)$-coresets suffices. Using recent results from [FSS13] and [GP14], the following theorem yields faster and smaller coreset constructions when $d \gg 1/\varepsilon$.

**Theorem 3.** *Let $P \subseteq \mathbb{R}^{d+1}$ and let $\varepsilon > 0$. A $(1, \varepsilon)$-coreset $C \subseteq \mathbb{R}^{d+1}$ for $P$ of size $|C| = O(1/\varepsilon^2)$ can be computed in $O(nd/\varepsilon^4)$ time.*

*Proof.* It was proven in [FSS13] that a coreset for $P$ and a family of query shapes, where each shape is spanned by $O(1)$ vectors in $\mathbb{R}^d$, can be computed by projecting $P$ on a $(1/\varepsilon^2)$ dimensional subspace $S$ that minimizes the sum of squared distances to $P$ up to a $(1 + \varepsilon)$ factor. The resulting coreset approximates the sum of squared distances to every such shape up to a factor of $(1 + \varepsilon)$. The size of this coreset is $n$, the same as the input size, however the coreset is contained in an $O(1/\varepsilon^2)$ dimensional subspace. We then compute a $(1, 0)$-coreset $C$ for this low dimensional set of $n$ points in $s = O(1/\varepsilon^2)$ space using Claim 2. This will take additional $O(ns^2)$ time and the resulting coreset will be of size $O(s)$.

The subspace $S$ can be computed deterministically in $O(nd/\varepsilon^4)$ using a recent result of [GP14]. $\square$

As proven below, the 1-segment mean of $C$ is an approximation to the 1-segment mean of $P$. So, using $C$ we can compute a fast approximation for the 1-segment mean of $P$.

**Corollary 4.** *Let $\varepsilon \in (0, 1)$. A $(1 + \varepsilon)$-approximation to the 1-segment of $P$ can be computed in $O(nd/\varepsilon^4)$ time.*

*Proof.* Using Theorem 3 we compute a $(1, \varepsilon)$-coreset $C$ of size $|C| = O(1/\varepsilon^2)$ in $O(nd/\varepsilon^4)$ time. Then, using the singular value decomposition it is easy to compute a 1-segment mean $f$ of $C$ in $O(d \cdot |C|^2) = O(d/\varepsilon^4)$ time. Hence, the overall running time is $O(nd/\varepsilon^4)$.

Let $f^*$ be a 1-segment mean of $P$ and $f$ be an arbitrary 1-segment. Since $C$ is a $(1, \varepsilon)$-coreset for $P$,

$$\text{cost}(P, f) \leq (1 + \varepsilon)\text{cost}(C, f) \leq (1 + \varepsilon)\text{cost}(C, f^*) \leq (1 + \varepsilon)^2\text{cost}(P, f^*) \leq (1 + 3\varepsilon)\text{cost}(P, f^*),$$

where in the last inequality we use the assumption $\varepsilon < 1$. Replacing $\varepsilon$ with $\varepsilon/3$ in the above proof proves the corollary. $\qquad\square$

# 4   No coreset $C \subseteq \mathbb{R}^{d+1}$ for $k \geq 3$

In the previous section we showed that a 1-segment coreset $(C, w)$ of size independent of $n$ exists for every signal $P$. Unfortunately, the next example shows that, in general, for $k \geq 3$ such a coreset $C$ must contain all the $n$ points of $P$. This result justifies the more complicated definition of a $(k, \varepsilon)$-coreset in the next section; See Definition 6.

**Claim 5.** *For every integers $n, c, d \geq 1$ there is a set $P$ of $n$ points in $\mathbb{R}^{d+1}$ such that the following holds. If $C \subseteq \mathbb{R}^{d+1}$ and $|C| < n$ then there is a 3-segment $f$ such that either*

$$\mathrm{cost}(C, f) \geq c \cdot \mathrm{cost}(P, f) \quad or \quad \mathrm{cost}(P, f) \geq c \cdot \mathrm{cost}(C, f).$$

*Proof.* Let $P = \{(i, 0, \cdots, 0)\}_{i=1}^{n}$, a constant-0 signal. Consider the 3-segment $f : \mathbb{R} \to \mathbb{R}^d$ such that $f(t) = (0, \cdots, 0)$ for every $t \in \mathbb{R}$. We have $\mathrm{cost}(P, f) = 0$. If $\mathrm{cost}(C, f) > 0$ then $\mathrm{cost}(C, f) \geq c \cdot \mathrm{cost}(P, f)$ as desired.

Otherwise, $\mathrm{cost}(C, f) = 0$. Let $(t, p) \in P \setminus C$ and consider a 3-segment $g : \mathbb{R} \to \mathbb{R}^d$ such that $g(t) = f(t) = (0, \cdots, 0)$ for every $t \in \mathbb{R} \setminus \{t\}$ and $g(t) \neq p$. Hence,

$$\begin{aligned}
\mathrm{cost}(C, g) &= \sum_{(t', p') \in C} \|p' - g(t')\|^2 = \sum_{(t', p') \in C \setminus (t, p)} \|p' - g(t')\|^2 \\
&= \sum_{(t', p') \in C} \|p' - f(t')\|^2 = \mathrm{cost}(C, f) = 0.
\end{aligned}$$

Since $\mathrm{cost}(P, g) = \|p - g(t)\|^2 > 0$ the last two inequalities imply $\mathrm{cost}(P, g) \geq c \cdot \mathrm{cost}(C, g)$. $\qquad\square$

# 5   Balanced Partition

A $(k, \varepsilon)$-coreset $D$ for a set $P$ approximates the fitting cost of a any query $k$-segment to $P$ up to a small multiplicative error of $1 \pm \varepsilon$. However, as proved in the previous section, such a coreset cannot be just a weighted subset of $\mathbb{R}^{d+1}$. Instead, we define a more involved data structure $D$ that represents the coreset, and define a new cost function $\mathrm{cost}'(D, f)$ that approximates the cost of $P$ to a $k$-segment $f$. We also assume that the time (first coordinate) is discrete between 1 to $n$. This means that the projecting of $P$ on any line can be described exactly in $O(d)$ space using only the first and last projected point, which motives the following structure of $D$.

The set $D$ consists of tuples of the type $(C, g, b, e)$. Each tuple corresponds to a different time interval $[b, e]$ in $\mathbb{R}$ and represents the set $P(b, e)$ of points of $P$ in this interval. The set $C$ is a $(1, \varepsilon)$-coreset for $P(b, e)$. Our first observation is that if all the points of the $k$-segment $f$ are on the same segment in this time interval, i.e, $\{f(t) \mid b \leq t \leq e\}$ is a linear segment, then the cost from $P(b, e)$ to $f$ can be approximated well by $C$, up to $(1 + \varepsilon)$ multiplicative error. We refer to these tuples as *coreset segments* in the description of the algorithm.

The second observation is that if we project the points of $P(b, e)$ on their 1-segment mean $g$, then the projected set $L$ of points will approximate well the cost of $P(b, e)$ to $f$, even if $f$ corresponds to more than one segment in the time interval $[b, e]$. Unlike the previous case, the error here is

additive. However, the third observation is that, since $f$ is a $k$-segment there will be at most $k-1$ time intervals $[b, e]$ that will intersects more than two segments of $f$. This motivates the following definition of $D$ and cost'.

**Definition 6** (cost'$(D, f)$)**.** *Let* $D = \{(C_1, g_1, b_1, e_1), (C_2, g_2, b_2, e_2), \cdots, (C_m, g_m, b_m, e_m)\}$ *where for every* $i \in [m]$ *we have* $C_i \subseteq \mathbb{R}^{d+1}$, $g_i : \mathbb{R} \to \mathbb{R}^d$ *and* $b_i \leq e_i \in \mathbb{R}$. *For a $k$-segment* $f : \mathbb{R} \to \mathbb{R}^d$ *and* $i \in [m]$ *we say that* $C_i$ *is* served by one segment *of $f$ if* $\{f(t) \mid b_i \leq t \leq e_i\}$ *is a linear segment. We denote by* $\mathrm{Good}(D, f) \subseteq [m]$ *the union of indexes $i$ such that $C_i$ is served by one segment of $f$. We also define* $L_i = \{g_i(\lceil t \rceil) \mid b_i \leq t \leq e_i\}$, *the projection of $C_i$ on $g_i$.*

*Finally, we define* cost'$(D, f)$ *to be*

$$\mathrm{cost}'(D, f) = \sum_{i \in \mathrm{Good}(D,f)} \mathrm{cost}(C_i, f) + \sum_{i \in [m] \setminus \mathrm{Good}(D,f)} \mathrm{cost}(L_i, f).$$

We will compute such a small structure $D$ that approximates cost$(P, f)$ for every $k$-segment $f$ using the above definition of cost'$(D, f)$. Such a set $D$ will be called a $(k, \varepsilon)$-coreset as follows.

**Definition 7** ($(k, \varepsilon)$-coreset)**.** *Let* $P \subseteq \mathbb{R}^{d+1}$, $k \geq 1$ *be an integer, and let* $\varepsilon \in (0, 1/10)$. *The set $D$ is a* $(k, \varepsilon)$-coreset *for $P$ if for every $k$-segment $f$ we have*

$$(1 - \varepsilon)\mathrm{cost}(P, f) \leq \mathrm{cost}'(D, f) \leq (1 + \varepsilon)\mathrm{cost}(P, f).$$

Our coreset construction is based on an input parameter $\sigma > 0$ such that for an appropriate $\sigma$ the output is a $(k, \varepsilon)$-coreset. Recall that for $\alpha, \beta > 0$, an $(\alpha, \beta)$-approximation for the $k$-segment mean of $P$ is a $(k \cdot \beta)$-segment $g$ such that cost$(P, g) \leq \alpha \cdot \mathrm{cost}(P, f^*)$. We show that using the value cost$(P, g)$ of such an approximation, even without knowing $g$, suffices to get a $(k, \varepsilon)$-coreset. In the next section we will compute such an $(\alpha, \beta)$-approximation for small $\alpha$ and $\beta$.

The size of the resulting coreset depends on $\alpha$ and $\beta$. In particular, for $\alpha = \beta = 1$ the following lemma implies that there exists a $(k, \varepsilon)$-coreset of size $O(k/\varepsilon^2)$ for every input set $P$.

---

**Algorithm 2:** BALANCEDPARTITION$(P, \varepsilon, \sigma)$.

**Input**: A set $P = \{(1, p_1), \cdots, (n, p_n)\}$ in $\mathbb{R}^{d+1}$
an error parameters $\varepsilon \in (0, 1/10)$ and $\sigma > 0$.
**Output**: A set $D$ that satisfies Lemma 8.

1   $Q := \emptyset$; $D = \emptyset$ ;
2   $p_{n+1} :=$ an arbitrary point in $\mathbb{R}^d$ ;
3   **for** $i := 1$ *to* $n + 1$ **do**
4      $Q := Q \cup \{(i, p_i)\}$;
5      $f^* :=$ a 2-approximation to the 1-segment mean of $Q$.   /* See Corollary 4         */
6      $\lambda := \mathrm{cost}(Q, f^*)$ ;
7      **if** $\lambda > \sigma$ **or** $i = n + 1$ **then**
8          $T := Q \setminus \{(i, p_i)\}$  /* Define the new coreset segment data up to $i$      */
9          $C :=$ a $(1, \varepsilon/4)$-coreset for $T$  /* See Claim 2                    */
10         $g :=$ a 2-approximation to the 1-segment mean of $T$  /* See Corollary 4    */
11         $b := i - |T|$;
12         $e := i - 1$;
13         $D := D \cup \{(C, g, b, e)\}$ /* Add a new coreset segment               */
14         $Q := \{(i, p_i)\}$ /* Start aggregating a new coreset segment          */
15   **return** $D$

---

**Lemma 8.** *Let* $P = \{(1, p_1), \cdots, (n, p_n)\}$ *such that* $p_i \in \mathbb{R}^d$ *for every* $i \in [n]$. *Suppose that* $h : \mathbb{R} \to \mathbb{R}^d$ *is an* $(\alpha, \beta)$-*approximation for the* $k$-*segment mean of* $P$, *and let*

$$\sigma = \frac{\varepsilon^2 \text{cost}(P, h)}{100k\alpha}.$$

*Let* $D$ *be the output of a call to* BALANCEDPARTITION$(P, \varepsilon, \sigma)$; *See Algorithm 2.*
*Then* $D$ *is a* $(k, \varepsilon)$-*coreset for* $P$ *of size*

$$|D| = O(k) \cdot \left( \frac{\alpha}{\varepsilon^2} + \beta \right),$$

*and can be computed in* $O(dn/\varepsilon^4)$ *time.*

*Proof.* Let $m = |D|$ and $f$ be a $k$-segment. We denote the $i$th coreset segment in $D$ by $(C_i, g_i, b_i, e_i)$ for every $i \in [m]$. For every $i \in [m]$ we have that $C_i$ is a $(1, \varepsilon/4)$-coreset for a corresponding subset $T = T_i$ of $P$. By the construction of $D$ we also have $P = T_1 \cup \cdots \cup T_m$.

Using Definition 6 of $\text{cost}'(D, f)$, $\text{Good}(D, f)$ and $L_i$, we thus have

$$|\text{cost}(P, f) - \text{cost}'(D, f)|$$

$$= |\sum_{i=1}^{m} \text{cost}(T_i, f) - \left( \sum_{i \in \text{Good}(D, f)} \text{cost}(C_i, f) + \sum_{i \in [m] \backslash \text{Good}(D, f)} \text{cost}(L_i, f) \right)|$$

$$= | \sum_{i \in \text{Good}(D, f)} (\text{cost}(T_i, f) - \text{cost}(C_i, f)) + \sum_{i \in [m] \backslash \text{Good}(D, f)} (\text{cost}(T_i, f) - \text{cost}(L_i, f)) \qquad (2)$$

$$\leq \sum_{i \in \text{Good}(D, f)} |\text{cost}(T_i, f) - \text{cost}(C_i, f)| + \sum_{i \in [m] \backslash \text{Good}(D, f)} |\text{cost}(T_i, f) - \text{cost}(L_i, f)|,$$

where the last inequality is due to the triangle inequality. We now bound each term in the right hand side.

For every $i \in \text{Good}(D, f)$ we have that $C_i$ is a $(1, \varepsilon/4)$-coreset for $T_i$, so

$$|\text{cost}(T_i, f) - \text{cost}(C_i, f)| \leq \frac{\varepsilon \text{cost}(T_i, f)}{4}. \qquad (3)$$

For every $i \in [m] \backslash \text{Good}(D, f)$, we have

$$|\text{cost}(T_i, f) - \text{cost}(L_i, f)| = \left| \sum_{(p,t) \in T_i} \|p - f(t)\|^2 - \sum_{t=b_i}^{e_i} \|g_i(t) - f(t)\|^2 \right|$$

$$= \left| \sum_{(p,t) \in T_i} \left( \|p - f(t)\|^2 - \|g_i(t) - f(t)\|^2 \right) \right| \qquad (4)$$

$$\leq \sum_{(p,t) \in T_i} \left| \|p - f(t)\|^2 - \|g_i(t) - f(t)\|^2 \right| \qquad (5)$$

$$\leq \sum_{(p,t) \in T_i} \left( \frac{12 \|g_i(t) - p\|^2}{\varepsilon} + \frac{\varepsilon \|p - f(t)\|^2}{2} \right) \qquad (6)$$

$$= \frac{12 \text{cost}(T_i, g_i)}{\varepsilon} + \frac{\varepsilon \text{cost}(T_i, f)}{2} \leq \frac{24\sigma}{\varepsilon} + \frac{\varepsilon \text{cost}(T_i, f)}{2}, \qquad (7)$$

where (5) is by the triangle inequality, and (6) is by the weak triangle inequality (see [FSS13, Lemma 7.1]). The inequality in (7) is because by construction $\mathrm{cost}(T, f^*) \le \sigma$ for some 2-approximation $f^*$ of the 1-segment mean of $T$. Hence, $\mathrm{cost}(T, g_i) \le 2\mathrm{cost}(T, f^*) \le 2\sigma$.

Plugging (7) and (3) in (2) yields

$$|\mathrm{cost}(P, f) - \mathrm{cost}'(D, f)| \le \sum_{i \in \mathrm{Good}(D,f)} \frac{\varepsilon \mathrm{cost}(T_i, f)}{4} + \sum_{i \in [m] \setminus \mathrm{Good}(D,f)} \left( \frac{24\sigma}{\varepsilon} + \frac{\varepsilon}{2} \mathrm{cost}(T_i, f) \right)$$

$$\le \left( \frac{\varepsilon}{4} + \frac{\varepsilon}{2} \right) \mathrm{cost}(P, f) + \frac{24k\sigma}{\varepsilon},$$

where in the last inequality we used that fact that $|[m] \setminus \mathrm{Good}(D, f)| \le k - 1 < k$ since $f$ is a $k$-segment. Substituting $\sigma$ yields

$$|\mathrm{cost}(P, f) - \mathrm{cost}'(D, f)| \le \frac{3\varepsilon}{4} \mathrm{cost}(P, f) + \frac{\varepsilon \mathrm{cost}(P, h)}{4\alpha} \le \frac{3\varepsilon}{4} \mathrm{cost}(P, f) + \frac{\varepsilon \mathrm{cost}(P, f)}{4} = \varepsilon \mathrm{cost}(P, f).$$

**Bound on $|D|$:** Let $j \in [m - 1]$, consider the values of $T$, $Q$ and $\lambda$ during the execution of Line 8 when $T = T_j$ is constructed. Let $Q_j = Q$ and $\lambda_j = \lambda$. The cost of the 1-segment mean of $Q_j$ is at least $\lambda_j/2 > \sigma/2 > 0$, which implies that $|Q_j| \ge 3$ and thus $|T_j| \ge 1$. Since $Q_{j-1}$ is the union of $T_{j-1}$ with the first point of $T_j$ we have $Q_{j-1} \subseteq T_{j-1} \cup T_j$. By letting $g^*$ denote a 1-segment mean of $T_{j-1} \cup T_j$ we have

$$\mathrm{cost}(T_{j-1} \cup T_j, g^*) \ge \mathrm{cost}(Q_{j-1}, g^*) \ge \lambda_j/2 > \sigma/2.$$

Suppose that for our choice of $j \in [m-1]$, the points in $T_{j-1} \cup T_j$ are served by a single segment of $h$, i.e, $\{h(t) \mid b_{j-1} \le t \le e_j\}$ is a linear segment. Then

$$\mathrm{cost}(T_{j-1}, h) + \mathrm{cost}(T_j, h) = \mathrm{cost}(T_{j-1} \cup T_j, h) \ge \mathrm{cost}(T_{j-1} \cup T_j, g^*) > \sigma/2. \tag{8}$$

Let $G \subseteq [m - 1]$ denote the union over all values $j \in [m - 1]$ such that $j$ is both even and satisfies (8). Summing (8) over $G$ yields

$$\mathrm{cost}(P, h) = \sum_{j \in [m]} \mathrm{cost}(T_i, h) \ge \sum_{j \in G} (\mathrm{cost}(T_{j-1}, h) + \mathrm{cost}(T_j, h)) \ge |G|\sigma/2. \tag{9}$$

Since $h$ is a $(\beta k)$-segment, at most $(\beta k) - 1$ sets among $T_1, \cdots, T_m$ are not served by a single segment of $h$, so $|G| \ge (m - \beta k)/2$. Plugging this in (9) yields $\mathrm{cost}(P, h) \ge (m - \beta k)\sigma/4$. Rearranging,

$$m \le \frac{4\mathrm{cost}(P, h)}{\sigma} + \beta k = O\left( \frac{k\alpha}{\varepsilon^2} \right) + \beta k. \tag{10}$$

**Running time:** In Theorem 3 it was shown how to compute a $(1, \varepsilon)$-coreset $C$ in time $O(dn/\varepsilon^4)$ for $n$ points using the algorithm in [GP14]. This algorithm is dynamic and supports insertion of a new point in $O(d/\varepsilon^4)$ time. Therefore, updating the 1-segment mean $f^*$ and the coreset $C$ can be done in $O(d/\varepsilon^4)$ time per point, and the overall running time is $O(nd/\varepsilon^4)$. $\qquad\square$

# 6 $(\alpha, \beta)$-Approximation

---

**Algorithm 3:** BICRITERIA$(P, k)$

---

**Input**: A set $P \subseteq \mathbb{R}^{d+1}$ and an integer $k \geq 1$
**Output**: An $(O(\log n), O(\log n))$-approximation to the $k$-segment mean of $P$.

1 **if** $n \leq 2k + 1$ **then**
2 $\quad$ $f :=$ a 1-segment mean of $P$;
3 $\quad$ **return** $f$;
4 Set $t_1 \leq \cdots \leq t_n$ and $p_1, \cdots, p_n \in \mathbb{R}^d$ such that $P = \{(t_1, p_1), \cdots, (t_n, p_n)\}$
5 $m \leftarrow \{t \in \mathbb{R} \mid (t, p) \in P\}$
6 Partition $P$ into $4k$ sets $P_1, \cdots, P_{2k} \subseteq P$ such that for every $i \in [2k - 1]$:

$$(i) \ | \{t \mid (t, p) \in P_i\} | = \left\lfloor \frac{m}{4k} \right\rfloor, \text{ and}$$

$$(ii) \ \text{if } (t, p) \in P_i \text{ and } (t', p') \in P_{i+1} \text{ then } t < t'.$$

7 **for** $i := 1$ *to* $4k$ **do**
8 $\quad$ Compute a 2-approximation $g_i$ to the 1-segment mean of $P_i$
9 $Q :=$ the union of $k + 1$ signals $P_i$ with the smallest value $\text{cost}(P_i, g_i)$ among $i \in [2k]$.
10 $h := $ BICRITERIA$(P \setminus Q, k)$
11 Set

$$f(t) := \begin{cases} g_i(t) & \exists (t, p) \in P_i \text{ such that } P_i \subseteq Q \\ h(t) & \text{otherwise} \end{cases}.$$

12 **return** $f$;

---

**Theorem 9.** *Let $f : \mathbb{R} \to \mathbb{R}^d$ be the output of a call to* BICRITERIA$(P, k)$*. Then*

*(i) $f$ is a $(\beta k)$-segment for some $\beta = O(\log n)$.*

*(ii) $\text{cost}(P, f) \leq \alpha \text{cost}(P, f^*)$, where $\alpha = \log_2 n$, and $f^*$ is a $k$-segment mean of $P$.*

*(iii) $f$ can be computed in $O(dn)$ time.*

*Proof.* *(i)* In every recursive iteration of the algorithm we remove $(k - 1)$ subsets of $P$, whose overall size is

$$|Q| \geq (k - 2) \cdot \left\lfloor \frac{n}{2k} \right\rfloor \geq (k - 2) \cdot \left( \frac{n}{2k} - 1 \right) = \frac{n}{2} - \frac{n}{k} - (k - 2) \geq \frac{n}{2} - \frac{n}{3} - \frac{n}{12} = \frac{n}{12},$$

where in the last inequality we used the assumption $k \in [3, n/12]$. Hence, the size of $P$ reduced by a constant fraction in each recursive iteration and we have $O(\log n)$ iterations.

Each subset $P_i$ in $Q$ contributes at most 2 segments to $f$, so the number of segments in $f$ increases by $O(k)$ in each of the $O(\log n)$ recursive iterations. Hence, the final output $f$ has $O(k \log n)$ segments.

*(ii)* Consider the value of $P$ during one of the recursive iterations. Since $f^*$ is a $k$-segment, every set in $P_1, \cdots, P_{2k}$ is served by one segment of $f^*$, except at most $k - 1$ such subsets. Let $M \subseteq [2k]$ denote the indexes of these (at most $k - 1$) subsets, and let $W = [2k] \setminus M$ denote the

rest, such that $Q = \bigcup_{i \in W} P_i$. Hence,

$$\text{cost}(P, f^*) \geq \sum_{i \in W} \text{cost}(P_i, f^*) \geq \sum_{i \in W} \min_g \text{cost}(P_i, g) \geq \frac{1}{2} \sum_{i \in [2k] \setminus M} \text{cost}(P_i, g_i), \qquad (11)$$

where the minimum is over every 1-segment $g : \mathbb{R} \to \mathbb{R}^d$. Since

$$|[2k] \setminus M| = 2k - |M| \geq 2k - (k-1) = k+1,$$

we have

$$\sum_{i \in [2k] \setminus M} \text{cost}(P_i, g_i) \geq \sum_{i \in W} \text{cost}(P_i, g_i) = \sum_{i \in W} \text{cost}(P_i, f) = \text{cost}(Q, f).$$

Plugging the last inequality in (11) yields $\text{cost}(Q, f) \leq 2\text{cost}(P, f^*)$. Summing over all iterations proves the claim.

*(iii)* In each recursive iteration, the dominated running time is in the "for" loop in Lines 7–8. We compute a 2-approximation $g_i$ for the 1-segment mean of a set $P_i$ of $m$ points in $O(md)$ time using Corollary 4. Hence, the overall time to compute Lines 7–8 is $O(nd)$. Since the size of $P$ reduced by a constant fraction in each recursive iteration, the overall running time is dominated by the first iteration which takes $O(nd)$ time. □

# 7 $(k, \varepsilon)$-Coreset

We now define the $k$-segment coreset, present a coreset construction algorithm, prove bounds on how well the coreset represents data with respect to the fitting cost to a $k$-segment query, and establish the running time complexity of the algorithm.

---

**Algorithm 4:** CORESET$(P, k, \varepsilon)$

---

    **Input**: A set $P = \{(1, p_1), \cdots, (n, p_n)\}$ in $\mathbb{R}^{d+1}$ .
    **Output**: A $(k, \varepsilon)$-coreset $(C, w)$ that satisfies Theorem 10.
**1** Compute $h \leftarrow$ BICRITERIA$(P, k)$ ; See Algorithm 3
**2** Set $\sigma \leftarrow \frac{\varepsilon^2 \text{cost}(P, h)}{100k \log_2 n}$
**3** Set $D \leftarrow$ BALANCEDPARTITION$(P, \varepsilon, \sigma)$ ; See Algorithm 2
**4** **return** $D$

---

**Theorem 10.** *Let $P = \{(1, p_1), \cdots, (n, p_n)\}$ such that $p_i \in \mathbb{R}^d$ for every $i \in [n]$. Let $D$ be the output of a call to CORESET$(P, k, \varepsilon)$; see Algorithm 2.*
*Then $D$ is a $(k, \varepsilon)$-coreset for $P$ of size*

$$|D| = O(k) \cdot \left( \frac{\log n}{\varepsilon^2} \right),$$

*and can be computed in $O(dn/\varepsilon^4)$ time.*

*Proof.* By Theorem 9, $h$ is an $(\alpha, \beta)$-approximation for the $k$-segment mean of $P$ for $\alpha = \log_2 n$ and $\beta = O(\log n)$. Theorem 10 then follows by substituting $\alpha$ and $\beta$ in Theorem 8. □

# 8 Weak $(k, \epsilon)$ Coreset for Efficient Segmentation

When computing an optimal k-segmentation for our data, we are bounded by the scale of the data in yet another aspect – the number of possible locations for each segment endpoint is $O(N)$. This means we cannot run algorithms with a linear complexity in the data size, let alone a quadratic one, as the original method of [Bel61]. While the coreset we propose handles gracefully $k$-segmentations whose endpoints lie on the coreset segments boundaries, it does not handle the more general case, where we want endpoints that do not coincide with out coreset segment endpoints. In the endpoint limited case, we use the same dynamic programming framework suggested by Bellman in [Bel61]. Since the number of possible segments endpoints is $O\left(\frac{(k\alpha)}{\epsilon^2} + \beta k\right)$, the number of steps in the algorithm is

$$O\left(\left(\frac{(k\alpha)}{\epsilon^2} + \beta k\right)^2 k\right). \tag{12}$$

For the more general case (without endpoint constraints), we now describe an additional new approximation tool in use by our algorithm when computing efficient k-segmentation. During the computation of an optimal segmentation, exhaustive search must be performed when updating the segmentation for the $k + 1$-segments induction step. Since an $O(N)$ operation such as this is too costly to compute, we require a way of approximating this search.

For an integer $n \geq 1$ we denote $[n] = \{1, \cdots, n\}$. Let $k, n \geq 1$ be a pair of integers. A function $f : \mathbb{R} \to [0, \infty)$ is *non-decreasing* over $[n]$ if $f(i) \leq f(j)$ for every $i \leq j$ in $[n]$, and *non-increasing* if $f(i) \geq f(j)$ for every $i \leq j$ in $[n]$. A function is *monotonic* if it is either non-increasing or non-decreasing. A function $g : \mathbb{R} \to [0, \infty)$ is *k-piecewise monotonic* if $[n]$ can be partitioned into $k$ consecutive sub-intervals $[n] = [i_1] \cup ([i_2] \setminus [i_1]) \cdots \cup ([n] \setminus [i_{k-1}])$ such that $g$ is monotonic over each one of them.

---

**Algorithm 5:** PIECEWISECORESET$(n, s, \varepsilon)$

**Input**: An integer $n \geq 1$ , a function $s : [n] \to (0, \infty)$ and an error parameter $\varepsilon > 0$.
**Output**: A vector $w = (w_1, \cdots, w_n)$ that satisfies Lemma 11.

**1** Set $t := \sum_{j=1}^{n} s_j$ and $B := \emptyset$ ;

**2 for** $i = 1$ *to* $n$ **do**

**3** $\quad$ Set $b_i := \left\lceil \dfrac{\sum_{j=1}^{i} s_i}{\varepsilon t} \right\rceil$ `'/* Hence, ` $b_i \leq \lceil 1/\varepsilon \rceil$ `*/`

**4** $\quad$ **if** $b_i \notin \{b_j \mid j \in B\}$ **then**

**5** $\quad\quad$ $| \quad B := B \cup \{i\}$

**6 for** *each* $j \in B$ **do**

**7** $\quad |$ Set $I_j := \{i \in [n] \mid b_i = b_j\}$;

**8** $w_j := \begin{cases} \frac{1}{s_j} \sum_{i \in I_j} s_i & j \in B \\ 0 & \text{otherwise.} \end{cases}$ ;

**9 return** $(w_1, \cdots, w_n)$

---

**Lemma 11.** *Let $k, n \geq 1$ be a pair of integers, $\varepsilon > 0$ and let $f, s : [n] \rightarrow (0, \infty)$ be a pair of $k$-piecewise monotonic functions. Let $w = (w_1, \cdots, w_n) \in \mathbb{R}^n$ denote the output of a call to* PIECEWISECORESET$(n, s, \varepsilon/(4k \sum_{i=1}^n s_i))$; *see Algorithm 5. If for every $i \in [n]$*

$$f(i) \leq s_i \sum_{j=1}^n f(j) \tag{13}$$

*then*

$$(1 - \varepsilon) \sum_{i=1}^n f(i) \leq \sum_{i=1}^n w_i f(i) \leq (1 + \varepsilon) \sum_{i=1}^n f(i).$$

*Proof.* For every $i \in [n]$ let

$$h_i = \frac{f(i)}{s_i \sum_{j=1}^n f(j)}.$$

We will prove that for a vector $w$ that is returned from a call to PIECEWISECORESET$(n, s, \varepsilon)$ we have

$$\left| \sum_i \frac{s_i}{t} \cdot h_i - \sum_{j \in B} \frac{w_j s_j}{t} \cdot h_j \right| \leq 4\varepsilon k, \tag{14}$$

Multiplying this by $t \sum_{i=1}^n f(i)$ yields

$$| \sum_{i=1}^n f(i) - \sum_{j \in B} w_j f(j) | = | \sum_{i=1}^n f(i) - \sum_{j=1}^n w_j f(j) | \leq 4kt\varepsilon \sum_i f(i).$$

Replacing $\varepsilon$ by $\varepsilon/(4kt)$ proves Lemma 11.

Since both $s$ and $f$ are $k$-piecewise monotonic, $h$ is $2k$-monotonic. Hence, there is a partition $\Pi = \{[i_1], [i_2] \setminus [i_1], \cdots, [n] \setminus [i_{2k-1}]\}$ of $[n]$ into consecutive $2k$ intervals such that $h_i$ is monotonic over each of these intervals.

Let $I_j = \{i \in [n] : b_i = b_j\}$ for every $j \in B$. For every $I \in \Pi$ we define $\text{Good}(I) = \{j \in B \mid I_j \subseteq I\}$. Their union is denoted by $\text{Good} = \sum_{I \in \Pi} \text{Good}(I)$. Hence,

$$\left| \sum_{i \in [n]} \frac{s_i}{t} \cdot h_i - \sum_{j \in B} \frac{w_j s_j}{t} \cdot h_j \right| = \left| \sum_{i \in [n]} \frac{s_i}{t} \cdot h_i - \sum_{j \in B} \frac{\sum_{i \in I_j} s_i}{t} \cdot h_j \right| = \sum_{j \in B} \sum_{i \in I_j} \frac{s_i}{t} \cdot (h_i - h_j)$$

$$\leq \left| \sum_{j \in B \setminus \text{Good}} \sum_{i \in I_j} \frac{s_i}{t} \cdot (h_i - h_j) \right| \tag{15}$$

$$+ \sum_{I \in \Pi} \left| \sum_{j \in \text{Good}(I)} \sum_{i \in I_j} \frac{s_i}{t} \cdot (h_i - h_j) \right|. \tag{16}$$

We now bound (15) and (16). Put $j \in B$. By Line 4 of the algorithm we have $|I_j \cap B| = 1$ and $\sum_{i \in I_j} s_i/t \leq \varepsilon$. Hence,

$$\left| \sum_{i \in I_j} \frac{s_i}{t} \cdot (h_i - h_j) \right| \leq \varepsilon(\max_{i \in I_j} h_i - \min_{i \in I_j} h_i) \leq \varepsilon, \tag{17}$$

where the last inequality holds since $h_i \leq 1$ for every $i \in [n]$, by (13). Since each set $I \in \Pi$ contains consecutive numbers, we have $|B \setminus \text{Good}| \leq 2k$. Using this and (17), we bound (15) by

$$\left| \sum_{j \in B \setminus \text{Good}} \sum_{i \in I_j} \frac{s_i}{t} \cdot (h_i - h_j) \right| \leq |B \setminus \text{Good}| \cdot \varepsilon \leq |\Pi| \varepsilon \leq 2\varepsilon k. \tag{18}$$

Put $I \in \Pi$ and denote the numbers in $\text{Good}(I)$ by $\text{Good}(I) = \{k, k+1, \cdots, \ell\}$. Recall that $h$ is monotonic on $I$. Without loss of generality, assume that $h$ is non-decreasing on $I$. Therefore, summing (17) over $\text{Good}(I)$ yields

$$| \sum_{j \in \text{Good}(I)} \sum_{i \in I_j} \frac{s_i}{t}(h_i - h_j)| \leq \sum_{j=k}^{\ell} \left| \sum_{i \in I_j} \frac{s_i}{t}(h_i - h_j) \right| \leq \sum_{j=k}^{\ell} \varepsilon(\max_{i \in I_j} h_i - \min_{i \in I_j} h_i)$$

$$\leq \varepsilon \sum_{j=k}^{\ell-1} (\min_{i \in I_{j+1}} h_i - \min_{i \in I_j} h_i) = \varepsilon(\min_{i \in I_\ell} h_i - \min_{i \in I_1} h_i) \leq \varepsilon,$$

where in the last derivation we used the fact that $h_i \leq 1$ for every $i \in [n]$. Summing over every $I \in \Pi$ bounds (16) as,

$$\sum_{I \in \Pi} \left| \sum_{j \in \text{Good}(I)} \sum_{i \in I_j} \frac{s_i}{t} \cdot (h_i - h_j) \right| \leq |\Pi| \cdot \varepsilon \leq 2\varepsilon k.$$

Plugging (18) and the last inequality in (15) and (16) respectively proves (14) as

$$\left| \sum_{i \in [n]} \frac{s_i}{t} \cdot h_i - \sum_{j \in B} \frac{w_j s_j}{t} h_j \right| \leq 4\varepsilon k$$

$\square$

For every $p, q \in \mathbb{R}^d$ we denote $D(p,q) = \|p - q\|^2$, where $\|p - q\|$ is the Euclidean distance between $p$ and $q$.

**Lemma 12.** *Let $p_1, \cdots, p_n$ be a set of points on a line in $\mathbb{R}^d$ such that $\|p_1 - p_2\| = \cdots = \|p_{n-1} - p_n\| = \Delta$ for some $\Delta \geq 0$ and the first coordinate of $p_i$ is $i$ for every $i \in [n]$. Let $\ell : \mathbb{R} \to \mathbb{R}^d$ be a function such that $\{(x, \ell(x)) \mid x \in \mathbb{R}\}$ is a line in $\mathbb{R}^{d+1}$. Then for every $i \in [n]$*

$$\|p_i - \ell(i)\|_2^2 \leq \frac{4 \sum_{j \in [i]} \|p_i - \ell(i)\|_2^2}{i}.$$

*Proof.* Since $P$ is contained in a line, it can be shown [FFS06] that there is a point $q \in \mathbb{R}^d$ and a positive number $w > 0$ such that for every $i \in [n]$

$$\|p_i - \ell(i)\|_2 = w\|p_i - q\|_2. \tag{19}$$

Let $\tilde{D} : [0, \infty) \to [0, \infty)$ be a monotone non-decreasing function and $r \in [0, \infty)$ such that $D(xe^\delta) \leq e^{r\delta}D(x)$ for every $x, \delta > 0$. It can be shown that for $\rho = \max\{2^{r-1}, 1\}$ and every $a, b, c \in M$ in a metric space $(M, \text{dist})$ we have

$$\tilde{D}(\text{dist}(a, c)) \leq \rho(\tilde{D}(\text{dist}(a, b)) + \tilde{D}(\text{dist}(b, c)));$$

See [FS12]. In particular, for the case $M = \mathbb{R}^d$, $\text{dist}(a, b) = w\|a - b\|$, we denote $D(a, b) = \tilde{D}(w\|a - b\|)$ to obtain

$$D(a, c) \leq \rho(D(a, b) + D(b, c)). \tag{20}$$

Let $m = \frac{1}{i} \sum_{j \in [i]} D(p_j, q)$ and $i \in [n]$. We will prove that

$$D(p_i, q) \leq 4m\rho^2. \tag{21}$$

In particular, for $\tilde{D}(x) = x^2$ we have $r = 2$, $\rho = 1$ and

$$\|p_i - \ell(i)\|_2^2 = \tilde{D}(\|p_i - \ell(i)\|) = \tilde{D}(w\|p_i - q\|) = D(p_i, q)$$

$$\leq 4m = \frac{4 \sum_{j \in [i]} \|p_i - \ell(i)\|_2^2}{i}, \tag{22}$$

where the second equality is by (19), and (22) is by (21).

Indeed, let $Q = \{j \in [i] \mid D(p_j, q) \leq 2m\}$. By Markov's inequality,

$$|Q| \geq \frac{i}{2}. \tag{23}$$

Hence, there are $p_s, p_t \in Q$ such that $s - t \geq i/2$. Using this and (20)

$$D(p_s, p_t) \leq \rho(D(p_s, q) + D(q, p_t)) \leq 2\rho m. \tag{24}$$

Since $s - t \geq i/2$,

$$\Delta\|p_i - p_s\| = \Delta(i - s) \leq \Delta(i - i/2) = \Delta i/2 \leq \Delta(s - t) = \Delta\|p_s - p_t\|.$$

Since $\tilde{D}$ is non-decreasing, the last equation implies $D(p_i, p_s) \leq D(p_s, p_t)$. Together with (24) we get $D(p_i, p_s) \leq 2\rho m$. Using the last inequality and the fact that $p_s \in Q$ proves (21) as

$$D(p_i, q) \leq \rho(D(p_i, p_s) + D(p_s, q)) \leq \rho(2\rho m + 2m) \leq 4m\rho^2.$$

$\square$

A function $g : \mathbb{R} \to \mathbb{R}^d$ is a 2-*piecewise linear function* if the set $\{(x, g(x)) \mid x \in \mathbb{R}\}$ is the union of two linear segments in $\mathbb{R}^{d+1}$.

**Corollary 13.** *Let* $(w_1, \cdots, w_n) \in \mathbb{R}^n$ *be the output of a call to* PIECEWISECORESET$(n, s, \frac{c\varepsilon}{\log n})$ *where* $c$ *is a sufficiently large universal constant,* $n \geq 1$, $\varepsilon > 0$ *and* $s$ *is the function that maps every* $i \in [n]$ *to* $s_i = \max\left\{\frac{4}{i}, \frac{4}{n-i+1}\right\}$.

*Then for every set* $(1, p_1), \cdots, (n, p_n)$ *of* $n$ *points that is contained in a line in* $\mathbb{R}^{d+1}$ *and every* 2-*piecewise linear function* $g : \mathbb{R} \to \mathbb{R}^d$ *the following hold:*

1. $w$ *has* $\|w\|_0 = O\left(\frac{\log n}{\varepsilon}\right)$ *non-zeroes entries.*

2. $w$ *can be computed in* $O(\log n) \cdot \|w\|_0$ *time and* $\|w\|_0$ *space.*

3.

$$(1 - \varepsilon) \sum_{i=1}^n \|g(i) - p_i\|^2 \leq \sum_{i=1}^n w_i^2\|g(i) - p_i\|^2 \leq (1 + \varepsilon) \sum_{i=1}^n \|g(i) - p_i\|^2.$$

*Proof.* (*i*) Put $\varepsilon' = c\varepsilon \log n$. By Line 8 of the algorithm, $\|w\|_0 = |B|$. Since $B$ consists of distinct integers $b_i \in [1, 1/\varepsilon' + 1]$ we have $\|w\|_0 = |B| = O(1/\varepsilon') = O(\log(n)/\varepsilon)$.

(*ii*) Since $b_i$ is monotonic over $i \in [n]$, we can use binary search on $[n]$ to compute the smallest $i \in [n]$ such that $b_i \notin B$. In each of the $O(\log n)$ iterations we compute $b_j$ for some $j \in [n]$. Since $\sum_{j=1}^{i} s_i$ is a sum of two harmonic series, $b_j$ can be computed in $O(1)$ time. As explained in (i), $|B| = O(\log(n)/\varepsilon)$ so the overall time is $O(\log(n)/\varepsilon') = O(\log^2 n/\varepsilon)$. We only need to store $w$ during this recursion, which takes $\|w\|_0$ space.

(*iii*) Put $i \in [n]$ and let $f(i) = \|p_i - g(i)\|^2$. Since $(1, p_1), \cdots, (n, p_n)$ are on a line, we have that $\|p_1 - p_2\| = \cdots = \|p_{n-1} - p_n\| = \Delta$ for some $\Delta \geq 0$. Since $g$ is 2-piecewise linear function, there is a line $\{x, \ell(x)\}$ for some $\ell : \mathbb{R} \to \mathbb{R}^d$ such that $\ell(j) = g(j)$ for every $j \in [i]$ or every $j \in \{i, i+1, \cdots, n\}$. Without loss of generality, we assume the first case. By Lemma 12,

$$f(i) = \|p_i - g(i)\|_2^2 = \|p_i - \ell(i)\|_2^2 \leq \frac{4\sum_{j \in [i]} \|p_i - \ell(i)\|_2^2}{i} \leq s_i \sum_{j \in [i]} \|p_i - \ell(i)\|_2^2 = s_i \sum_{j \in [i]} f(j). \quad (25)$$

Since $g$ is 2-piecewise linear and $p_1, \cdots, p_n$ are points on a line, we have that $f$ is 4-monotonic over $[n]$. The function $s$ is 2-monotonic. We also have that

$$\frac{c\varepsilon}{\log n} \leq \frac{\varepsilon}{4k \sum_{i=1}^{n} s_i}$$

for a sufficiently small constant $c$. Plugging this and (25) in Lemma 11 then proves the theorem as

$$(1 - \varepsilon) \sum_{i=1}^{n} f(i) \leq \sum_{i=1}^{n} w_i f(i) \leq (1 + \varepsilon) \sum_{i=1}^{n} f(i).$$

$\square$

We show how to compute a $(1 + \varepsilon)$-approximation to the $k$-segment mean of the original signal $P$ using its coreset. The technique can be used to solve any other optimization problem over $k$-segments, assuming that we have an existing algorithm for a weighted signal. For example, if priors are given (weights for each segment) or we want to minimize the cost over some subset of $k$-segments (e.g., $(k, m)$-segment mean).

We assume that we are given a possibly inefficient algorithm SLOW-SEGMENTATION ("black box") that will be used to extract the will compute the $k$-segment mean of a small set that is based on the coreset. The algorithm SLOW-SEGMENTATION gets a set $Q$ of pairs $((t, p), w)$ where $t \in \mathbb{R}$, $(t, p)$ is a point in $\mathbb{R}^{d+1}$, and $w > 0$ denote its weight. The algorithm then returns the $k$-segment mean of $Q$, i.e., the $k$-piecewise linear function that minimizes the *weighted cost*

$$\mathrm{cost}_W(Q, f) := \sum_{((t,p),w) \in Q} w \| (p - f(t)) \|^2,$$

We will run this algorithm only on a small set $Q$, whose size is roughly the size of the coreset. In what follows we describe the algorithm FAST-SEGMENTATION that uses the coreset and SLOW-SEGMENTATION to get a fast approximation of the $k$-segment mean of the original set $P$.

**Algorithm overview**    The input to the algorithm FAST-SEGMENTATION is a signal $P$ of $n$ points in $\mathbb{R}^d$, an error parameter $\varepsilon > 0$, and an integer $k \geq 1$. In addition, the algorithm gets a pointer to the algorithm SLOW-SEGMENTATION.

---

**Algorithm 6:** FAST-SEGMENTATION($P, k, \varepsilon$, SLOW-SEGMENTATION)

**Input**:

- A set $P = \{(1, p_1), \cdots, (n, p_n)\}$ in $\mathbb{R}^{d+1}$,

- an integer $k \geq 1$,

- an error parameter $\varepsilon > 0$, and

- an algorithm SLOW-SEGMENTATION($Q, k$) that computes the $k$-segment mean of a given weighted set $Q$.

**Output**: A $(1 + \varepsilon)$-approximation $f$ to the $k$-segment mean of $P$.

**1** $D \leftarrow$ CORESET($P, k, \varepsilon$); See Algorithm 4.
**2** Identify $D = \{(C_1, g_1, b_1, e_1), .., (C_m, g_m, b_m, e_m)\}$
**3** $Q \leftarrow \emptyset$
**4** **for** $i \leftarrow 1$ *to* $m$ **do**
**5** $\quad$ $P_i \leftarrow \{(b_i, g_i(b_i)), \cdots, (e_i, g_i(e_i))\}$
**6** $\quad$ $(w_1, \cdots, w_n) \leftarrow$ PIECEWISECORESET($|P_i|, s, c\varepsilon / \log(n)$), where $c$ and $s$ are defined in
$\quad$ Corollary 13.
**7** $\quad$ $Q \leftarrow Q \cup \left\{(t, p), w_j^2) \mid (t, p) \text{ is the } j\text{th point of the signal } P_i\right\}$
**8** $h \leftarrow$ SLOW-SEGMENTATION($Q, k$)
**9** **for** $i \leftarrow 1$ *to* $m$ **do**
**10** $\quad$ $T_i \leftarrow \{b_i, \cdots, e_i\}$
**11** $\quad$ **if** $\{h(t) \mid t \in T_i\}$ *consists of at most 2-segments* **then**
**12** $\quad\quad$ $f(t) \leftarrow h(t)$ for every $t \in T_i$
**13** $\quad$ **else**
**14** $\quad\quad$ $f(t) \leftarrow g_i(t)$ for every $t \in T_i$
**15** $\quad$ **return** $f$

---

Recall that for a $k$-segment $f : \mathbb{R} \to \mathbb{R}^d$ and $i \in [m]$ we say that $C_i$ is served by one segment of $f$ if $\{f(t) \mid b_i \leq t \leq e_i\}$ is a linear segment. The next lemma states the weighted set $Q$ that is computed in Line 7 of Algorithm 6 is a weak coreset in the following sense. For every $k$-segment $f$ such that each cell $C_i$ is served by at most two segments of $f$, the cost of $P$ and the weighted cost of $Q$ to $f$ are approximately the same. In Theorem 9 we prove that a $k$-segment mean has this property, and thus can be computed from this coreset $Q$.

**Lemma 14** (Weak coreset). *Let $f$ be a $k$-segment such that $C_i$ is served by at most two segments of $f$, for every $i \in [m]$. Then*

$$\min_f \text{cost}(P, f) \leq \min_f \text{cost}_W(Q, f) \leq (1 + \varepsilon) \min_f \text{cost}(P, f).$$

*Proof.* Put $i \in [m]$ and $P_i = \left\{(t_1, p_1), \cdots, (t_{|P_i|}, p_{|P_i|})\right\}$. Since $C_i$ is served by at most two segments

of $f$, then $P_i$ is also served by at most two segments of $f$. By Corollary 13,

$$(1 - \varepsilon) \sum_{j=1}^{|P_i|} w_j \|f(t_j) - p_j\|^2 \leq \sum_{j=1}^{|P_i|} w_j^2 \|f(t_j) - p_j\|^2 \leq (1 + \varepsilon) \sum_{j=1}^{|P_i|} \|f(t_j) - p_j\|^2.$$

Hence, letting $Q_i = \left\{ (t_j, p_j), w_j^2) \mid w_j > 0, j \in [|P_i|] \right\}$, by Line 7 of Algorithm 6 we obtain

$$|\text{cost}(P_i, f) - \text{cost}_W(Q_i, f)| = \left| \sum_{j=1}^{|P_i|} \|f(t_j) - p_j\|^2 - \sum_{j=1}^{|P_i|} w_j^2 \|f(t_j) - p_j\|^2 \right|$$

$$\leq \varepsilon \sum_{j=1}^{|P_i|} \|f(t_j) - p_j\|^2 = \varepsilon \text{cost}(P_i, f).$$

Summing over every $i \in [m]$ yields

$$|\text{cost}(P, f) - \text{cost}_W(Q, f)| \leq \varepsilon \text{cost}(P, f).$$

$\square$

**Theorem 15.** *Let $P = \{(1, p_1), \cdots, (n, p_n)\}$ be a set in $\mathbb{R}^{d+1}$, $\varepsilon \in (0, 1/2)$, and $k \geq 1$ be an integer. Let $f : \mathbb{R} \to \mathbb{R}^d$ be the output of a call to* FAST-SEGMENTATION$(P, k, \varepsilon, \text{SegAlg})$. *Then $f$ is a $(1 + \varepsilon)$-approximation to the $k$-segment mean of $P$, i.e.,*

$$\text{cost}(P, f) \leq (1 + \varepsilon) \min_{f'} \text{cost}(P, f'),$$

*where the minimum is over every $k$-segment $g : \mathbb{R} \to \mathbb{R}^d$.*

*Proof.* Let $h$ be the $k$-segment that is computed in Line 8 of the algorithm FAST-SEGMENTATION$(P, k)$, and let $i \in [m]$. We first prove that $\text{cost}(P_i, f) \leq \text{cost}(P_i, h)$ by case analysis: (i) $f(t) = h_i(t)$ for every $t \in T_i$, and (ii) $f(t) = g_i(t)$ for every $t \in T_i$.
Case (i): In this case $\text{cost}(P_i, f) = \text{cost}(P_i, h)$ by definition of $P_i$.
Case (ii): In this case $\text{cost}(P_i, f) = \text{cost}(P_i, g_i)$. By its construction, $g_i$ is a 2-approximation for the 1-segment mean of $P_i$. Since the points of $P_i$ lie on a line, we thus have $\text{cost}(P_i, g_i) = 0$. Hence,

$$\text{cost}(P_i, f) = \text{cost}(P_i, g_i) = 0 \leq \text{cost}(P_i, h).$$

Summing $\text{cost}(P_i, f) \leq \text{cost}(P_i, h)$ over $i \in [m]$ yields

$$\text{cost}(P, f) \leq \text{cost}(P, h). \tag{26}$$

Suppose that $h^*$ minimizes $\text{cost}(P, f')$ over every $k$-segment $f' : \mathbb{R} \to \mathbb{R}^d$. Similarly to (26), it can be shown that there is a $k$-segment $f^*$ such that

$$\text{cost}(P, f^*) \leq \text{cost}(P, h^*),$$

and $C_i$ is served by at most two segments of $f^*$, for every $i \in [m]$. We then have

$$\text{cost}(P, f) \leq \frac{\text{cost}_W(Q, f)}{1 + \varepsilon} \tag{27}$$

$$\leq \frac{\text{cost}_W(Q, h)}{1 + \varepsilon} \tag{28}$$

$$\leq \frac{\text{cost}_W(Q, f^*)}{1 + \varepsilon} \tag{29}$$

$$\leq \frac{(1 - \varepsilon)\text{cost}(P, f^*)}{1 + \varepsilon} \tag{30}$$

$$\leq (1 + 10\varepsilon)\text{cost}(P, f^*), \tag{31}$$

where (28) holds by (26), Eq. (27) and (30) hold by Lemma 14, Eq. (29) is by the optimality of $h$, and (31) holds since $\varepsilon \leq 1/2$. Replacing $\varepsilon$ with $\varepsilon/10$ proves this theorem. $\qquad \square$

## 8.1 Efficient $k$-Segment Mean Computation using the coreset

However, since we cannot allow linear time search over the data, we add the additional constraints that there cannot be more than one $k$-segment endpoint inside each coreset-segment, and that each of the $k$-segments starts and terminates at an endpoint of the coreset segments. This allows us to use the coreset obtained by Algorithm 2 for cost computations coreset, and perform the computation of all linear segment costs required in [Bel61] on a sublinear number of sampling points, reducing overall algorithm complexity from $O(kN^2)$ to $O(k^3 log^2 N)$. By construction of the piecewise coresets, and the segments $C_i$ computed in Algorithm 2, the cost computed with these limitations on the endpoints is an $\epsilon$ approximation of the cost of our solution on the real data. Specifically, our solution is an $\epsilon$-approximation to the real optimal solution.

The modifications required compared to the algorithm of [Bel61] for this case are as follows

- During the search over $u_{k'}$, $u_{k'}$ is allowed only to be at locations which are part of the piecewise coreset of some segment in $D$.

- For each line segment $(u_{k'}, b)$, its fitting solution and cost is obtained by concatenating row-wise the matrices $C_i$ from each segment $i$ of $D$ completely contained inside $(u_{k'}, b)$, along with the sampling points inside $(u_{k'}, b)$ from partially contained segments of $D$, into a single matrix $C_{(u_{k'}, b)}$, and solving for the linear segment using $C_{(u_{k'}, b)}$.

- $h(u_{k'}, u_{k'})$ is defined to be infinite if two segment endpoints are inside a coreset segment.

The algorithm is described as Algorithm 7. Let $L_{coreset}$ denote the maximum number of inner-points per segment obtained from Algorithm 5. The number of segment fitting cost computations done is

$$O\left(\left(\frac{(k\alpha)}{\epsilon^2} + \beta k\right)^2 L_{coreset}^2 k\right) \tag{32}$$

**Theorem 16.** *Given a coreset $D$ as described in Algorithm 2, and a set of piecewise-coresets computed as in Algorithm 5 for each segment, Algorithm 7 finds an $\epsilon$-approximation of the $k$-segment mean in time $O(\text{polylog}(n)\,\text{poly}(k))$*

---
**Algorithm 7:** Solving for k-segmentation using a coreset
---
1: **for** $b = 1, 2, \ldots, m$ **do**
2:     Update the 1-segment solution for each subsegment starting at $t = 1$

$$f_1(b) = h(1, b)$$

3: **end for**
4: **for** $k' = 1, 2, \ldots, K$ **do**
5:     **for** $b = 1, 2, \ldots, m$ **do**
6:         **for** $u_{k'} = 1, 2, \ldots, b$ **do**
7:             Update the k'-segment solution by updating the cost $f_{k'}$ based on the $(k' - 1)$-segment solution $f_{k'-1}$

$$f_{k'}(b) = \min_{1 \geq u_{k'} \geq b} [h(u_{k'}, b) + f_{k'-1}(u_{k'})],$$

            where $h(u_{k'}, b)$ is computed using the appropriate matrix $C_{(u_{k'}, b)}$.
8:         **end for**
9:     **end for**
10: **end for**
---

*Proof.* Computation time is determined by the number of sampling points over the whole signal. Since each segment has $n$ points at most, we have $O(\frac{log n}{\epsilon})$ sampling points according to Corollary 13. Since there are $\left( \frac{(k\alpha)^2}{\epsilon^2} + \beta k \right)$ segments according to Equation 10, we have overall $O(\frac{log n}{\epsilon} \left( \frac{(k\alpha)}{\epsilon^2} + \beta k \right))$ sampling points. Therefore our algorithm requires $O \left( \left( \frac{log n}{\epsilon} \left( \frac{(k\alpha)}{\epsilon^2} + \beta k \right) \right) k \right)$ estimations of linear segment fittings. Each line segment estimation involves constructing a matrix composed out of $O \left( \frac{(k\alpha)}{\epsilon^2} + \beta k \right)$ complete segments plus possibly $O(\frac{log n}{\epsilon})$ sampling points, and inverting it. We note that each segment (partial or full) contributes $O(log n)$ rows to the matrix, and that its width is $O(d)$. Hence, inverting it is $O(\text{polylog}(n) \, \text{poly}(k))$, therefore the algorithm takes $O(\text{polylog}(n) \, \text{poly}(k))$ to complete. The approximation property of the algorithms comes from the approximation of the coresets $\qquad\qquad\qquad\qquad\qquad\qquad\qquad\qquad\qquad\qquad\qquad\qquad\qquad\square$

# 9   Parallel and Streaming Implementation

One major advantage of coresets is that they can be constructed in parallel as well as in a streaming setting. The main observation is that the union of coresets is a coreset — if a data set is split into subsets, and we compute a coreset for every subset, then the union of the coresets is a coreset of the whole data set. This allows us to have each machine separately compute a coreset for a part of the data, with a central node which approximately solves the optimization problem; see [FSS13, Theorem 10.1] for more details and a formal proof.

    When discussing streaming coresets, one must define the merging and reduction operations used in streaming, and show that the coresets create are still efficient and accurate. We build our merge and reduce operations as a modification of the coreset algorithm as given in Algorithm 4, so that it compacts coreset segments rather than signal points. For this we modify Algorithms 2,3 as we

now describe.

First, we look at Algorithm 3, we modify it in the following way. In line 6 of the algorithm, the original segments from both child coresets are taken. Partitioning is done by unifying existing coreset segments into sections $P_i$. Iterating over Theorem 9, we note that part i is kept by the reduction of parts at each turn. Looking at the proof of part ii we note that we only use the coreset segments' cost as represented for 1-segments, and this can be computed by the $C$ matrices, starting from the end of Equation 11. This holds also for the joined and compacted matrices.

Next, we look at Algorithm 2, and modify it to utilize the child coresets' coreset segments in order to construct a new set of coreset segments for the combined span. This requires several modification the the algorithm – notibly, the accumulation of a new coreset segment $Q$ is done solely in terms of adding new child coreset segments. We note that $f^*$ and $\lambda$ in lines 5 and 6 respectively can be computed for concatenations of coreset segments, in terms of their $(1, \varepsilon/4)$-coresets. We note that $C,g$ can be computed using the $C$ matrices of the child coresets. We do so by concatenating the $C$ child matrices, and recomputing the SVD for the concatenated matrix.

Specifically, let $(U_1, S_1, V_1)$ and $(U_2, S_2, V_2)$ be the SVD of matrices $P_1, P, 2$ corresponding to the coreset segments creation. It's easy to show that

$$\left\| (S_1 V_1^T) \begin{pmatrix} a \\ b \\ -I \end{pmatrix} \right\|_F^2 + \left\| (S_2 V_2^T) \begin{pmatrix} a \\ b \\ -I \end{pmatrix} \right\|_F^2 =$$
$$\left\| \begin{pmatrix} S_1 V_1^T \\ S_2 V_2^T \end{pmatrix} \begin{pmatrix} a \\ b \\ -I \end{pmatrix} \right\|_F^2 = \left\| U_J^T \begin{pmatrix} S_1 V_1^T \\ S_2 V_2^T \end{pmatrix} \begin{pmatrix} a \\ b \\ -I \end{pmatrix} \right\|_F^2 = \qquad (33)$$
$$\left\| S_J V_J^T \begin{pmatrix} a \\ b \\ -I \end{pmatrix} \right\|_F^2,$$

where $(U_J, S_J, V_J)$ is the SVD of $\begin{pmatrix} S_1 V_1^T \\ S_2 V_2^T \end{pmatrix}$, and we used the properties of the Frobenius norm, the isometry properties of unitary matrices, and properties of $(U_J, S_J, V_J)$, respectively. Once $C_J$ is computed $g_J$ can be computed easily.

Looking at the proof of Lemma 8, we note that the treatment of good coreset segments remains the same. The coreset segments that do not belong to $Good(D, f)$ still amount to the same cost bounds, due to the construction of $g_J$.