[Reviews · NeurIPS 2014]

Submitted by Assigned_Reviewer_1

The authors study the problem of k-segmentation of data and provide a compact coreset that is linear in size of the dimensions and number of segments.

Strengths:
1. Compact coreset that is computed in time linear in n, d, k.
2. The paper is well written and provides a good summarization of existing techniques.
3. The experiments include a collection of very interesting data sets.

Concerns:
(Note: A few concerns raised are directly in light of the work in [12], especially since the authors address [12] as the most recent paper worthy of comparison.)
1. In the abstract, the authors claim that the size of the coreset is independant of d and n, but in Page 3, the size of the coreset is given as O(dk/\eps^2) and in Page 5, the size of the coreset is given as O(klog(n)/\eps^2). It is therefore very unclear what the size of the coreset is.
2. The authors claim that the the size of their coreset is an improvement over [12] whose runtime is cited to be cubic in both d and k. But, the size of the coreset as claimed by [12] is linear in k.
3. The cost term involving the coreset is a little convoluted in the paper involing sum of two terms, whereas [12] provides a cleaner cost function very similar to the original cost function used on the entire data. This makes it difficult to evaluate the effectiveness of the coreset.
4. While the deterministic nature of the coreset computation is appealing, it is unclear how to run parallel and streaming versions of the algorithm since the effect of an adversary is scrambling the inputs each time is never discussed.
5. In its current form, the meat of the paper only has brief proof sketches and no analysis of run times making it contribution limited. The technical contribution of the paper is limited to a page since a flavor of BICRITERIA algorithm is already discussed in [12] (and not mentioned in this paper).
6. In page 7, the authors claim that they significantly outperform other schemes (Figure 2), while the other schemes discussed are merely uniform coreset and a random coreset (which take constant time to compute and are therefore much faster). The authors fail to discuss and compare their approach to existing state of the art methods including [12]. The methods that they compare to are from 40 years ago.
Summary: In its present form the paper is rather weak, although it provides a low space approximation to the k-segmentation problem. The authors should consider including more technical discussion including proofs and more detailed run time analysis to the body of the paper since the paper is about approximation.

Submitted by Assigned_Reviewer_11

In this paper, the authors provide a novel algorithm to compute the k-segmentation of streaming data. The problem is motivated by multiple applications from data mining, computer vision, biology, and finance. Given a signal P, which is a stream of points in Euclidean space, k-segmentation is to construct k linear functions, such that each linear function is an approximation of one interval of P. Roughly speaking, k-segmentation is to partition P into k pieces, and each piece is handled as a linear regression problem. There are several challenges on this problem. Firstly, how to divide P into k pieces? It is similar to clustering, but it is more complicated, since each piece should be an interval in the stream. Secondly, it should be computed in one pass with limited space complexity, since the input is a streaming data with large scale.

The paper presents a core-set based approach for this problem. The main idea is as follows. At first, it partitions P into 4k pieces, computes an approximation for each piece, and peels out the points with small error to the approximation. Then, it iteratively run this approximate-peel strategy, and output an (O(log n), O(log n))-approximation which is used for computing the core-set. Finally, in BlancedPartition-algorithm, it constructs the core-set via an incremental process, where each time adds a part of core-set which is a local representation of P.

There are several details the authors need concern:

1. In Line 209-210, it should make clear that what is g.

2. In Line 219, dose (O(log n), O(log n))-approximation mean a bi-criteria approximation?
Summary: Overall, I think this is an interesting paper, with enough technical innovations and important applications from real world. Particularly, it provides a theoretical guarantee on the size of core-set, which is a significant improvement comparing to previous results, like [12][15]. The experimental results are solid on multiple real datasets.

Submitted by Assigned_Reviewer_27

This paper presents a method for computing a k-segmentation of streaming temporal data using core-sets. The authors show that for the particular segmentation problem, it is possible to construct a core-set with size linear in k and better running time complexity than existing methods (most notably compared to related work from Feldman et al.)

Overall, I found the work interesting and well suited to the tasks described. The authors motivate the problem (and application to video summarization) well and provide clear description of the contributions and methods. I noticed there is a lot of connection to the work by Feldman et al., including the algorithms (for example, they had a Bicretria approximation and also motivate the streaming & parallel models). It will be useful to reference this in your method description and how your algorithm(s) differs.

However, some of the details in the experiments/analysis are glossed over (or missing) and should be clarified (e.g., “semantic” quality evaluation of the video frames). Also, in addition to the uniform segmentation (and other baselines listed) why isn’t there a comparison to [12]. Even though their method has worse running time & core-set size, perhaps this should have been another baseline system. It would also clearly show if/how your method outperforms in practice for the specific tasks.

Is there an explanation as to why there is a big gap between performance when using original vs. k-segment data in Fig. 5? It isn’t clear to me why DR on k-segment does so much better than DR on original data. Does this depend on the chosen method?
Summary: Overall the paper makes interesting contributions and shows improved runtime/memory complexity over existing methods. One suggestion would be to improve the presentation for the latter half of the paper by incorporating more details, analysis, etc.
Author Feedback
Author rebuttal: Dear Area Chair,
We thank the reviewers for their efforts and warm reviews, and look forward to incorporating their comments in the final version of the paper. The paper describes an end-to-end contribution (from theory to experiment). In the interest of space we moved many details to the appendix, which contains proofs for all the claims as well as algorithmic details.
It is important to note (cf. reviewer #1's comment) the contributions made in our algorithm compared to reference [12], (or other existing algorithms), as will be clarified in the final version. Specifically, our algorithm:
- Running time -- reduces the running time for computing the coreset to be linear in k and not cubic as in [12].
- Coreset elements count -- reduces the number of points in the coreset to be independent of d, instead of cubic in d as in [12].
- Coreset memory usage -- is the coreset elements count multiplied by d, as each d-dimensional point in the coreset takes O(d) memory.
- Coreset type -- uses significantly different (non-sampling) techniques than [12] in order to achieve the above results.

The improvement from algorithm that take time cubic in k and d -- to a totally new approach that takes linear time, allows us to compute the k-segmentation of high dimensional data (as video stream) or large number of segments (as llong GPS signals) in a practical and provable way for the first time. We thus believe that the result is more important than a usual incremental improvement of an existing algorithm..

We can further address the additional reviewer comments (in the order made) as we now describe:

Reviewer #1
1. The dependence on d is due to accessing each data element, as is often assumed in complexity analysis. The number of vector operations is still independent of d, i.e no additional complexity for larger d. We will clarify the notation in the final version. We will note the (log) dependence of |D| on n in the abstract.

2. Regarding time/space complexity w.r.t k, Construction time of the coreset in [12] is still cubic in k (the O(n) corollary 8.2 in [12] does not refer to k), and we improve upon this. The coreset size is linear in k, but there is no contradiction.

3. Regarding the cost comparison, in [12] there is also a separation of the cost into two sums (cf. [12], pg 6), later bounded by a single term. Theorem 4 in our paper gives a single-expression bound for the cost, w/ proof in the supplementary material.

4. Regarding adversary data ordering in streaming mode, our algorithm handles every possible adversarial stream with the same guarantees of time and quality; This behaviour of streaming coresets has been addressed, for example, in reference [10], Thm. 10.1, which refers to that point. We can note this point if it so pleases the reviewer.

5. Regarding full proofs in the paper -- in balancing between concrete empirical examples/datasets, introduction, and conveying the how/why of the algorithm, we have deferred significant parts of the analysis to the supplementary material. The high-level plan of the analysis proofs is kept, and the reader can both implement the algorithm and develop the proofs himself, or refer to the supplementary material for the full proofs.

6. Regarding the chosen comparison algorithms, the dependence on d in [12] would have made comparing to it infeasible for large d, and indeed that was one of the motivations for the new algorithm. The segmentation methods used (not the coreset) are the same as in [12], as are the subsampling alternatives.

Reviewer #11

1. In Line 209-210, it should make clear that what is g.
A: g is the 1-segment mean representation as mentioned in lines 204-205. We will make this explicit reference in the paper.

2. In Line 219, dose (O(log n), O(log n))-approximation mean a bi-criteria approximation?
A: yes, this is defined exactly in the supplementary material and we can explicitly note so in the paper.

Reviewer #27
1. Regarding comparison to Feldman et al. - in general, the framework is similar, but the specific algorithms (bicriteria, definitely the other parts) are different. More specifically, the complexity guarantees are different, for example w.r.t large d,k -- as this is a different algorithm. We can describe in more details in the final version of the paper.

2.Regarding semantics in video segmentation -- fair point, semantics in computer vision are not well defined, hence, we used the commonly accepted substitute of user judgment and made the comparison to human annotations. We can clarify this point in the final version.

3. On empirical comparison to [12] -- after experimenting with a code similar to [12], it is safe to say that the method from [12] is not competitive complexity-wise with the proposed method for high-dimensional. Hence, we did not include it.

4. On using DR with /without coresets -- the k-segmentation overcomes outliers in the data which tend to fail DR (which is L-infinity based). We noted this as well, and can relate to it in the paper.